# Drug repurposing screen identifies lonafarnib as respiratory syncytial virus fusion protein inhibitor

Respiratory syncytial virus (RSV) is a common cause of acute lower respiratory tract infection in infants, older adults and the immunocompromised. Effective directly acting antivirals are not yet available for clinical use. To address this, we screen the ReFRAME drug-repurposing library consisting of 12,000 small molecules against RSV. We identify 21 primary candidates including RSV F and N protein inhibitors, five HSP90 and four IMPDH inhibitors. We select lonafarnib, a licensed farnesyltransferase inhibitor, and phase III candidate for hepatitis delta virus (HDV) therapy, for further follow-up. Dose-response analyses and plaque assays confirm the antiviral activity (IC$_{50}$: 10-118 nM). Passaging of RSV with lonafarnib selects for phenotypic resistance and fixation of mutations in the RSV fusion protein (T335I and T400A). Lentiviral pseudotypes programmed with variant RSV fusion proteins confirm that lonafarnib inhibits RSV cell entry and that these mutations confer lonafarnib resistance. Surface plasmon resonance reveals RSV fusion protein binding of lonafarnib and co-crystallography identifies the lonafarnib binding site within RSV F. Oral administration of lonafarnib dose-dependently reduces RSV virus load in a murine infection model using female mice. Collectively, this work provides an overview of RSV drug repurposing candidates and establishes lonafarnib as a bona fide fusion protein inhibitor.

Acute RSV infection is the leading cause of severe lower-respiratory-tract infections in young children[1]. Moreover, the immunosuppressed and older adults are at risk of severe RSV infections. RSV is responsible for ca. 33 million episodes of airway infection in children younger than five years of age, leading to an estimated 3.2 million hospital admissions and 60,000 in-hospital deaths annually[2]. In the wake of the recent SARS-CoV-2 pandemic non-pharmacological interventions were in place to limit virus spread. These include the closing of schools and child care facilities and various measures of social distancing, which were implemented across the globe. Consequently, the epidemiology of other pathogens including RSV has been altered[3–7]. In case of RSV a transient suppression and subsequent resurgence of circulation has been documented in different countries. Parallel to this, a decline of RSV-specific antibodies was reported across all age-groups during the pandemic[8]. Although it is not clear how this decline correlates with susceptibility and antibody kinetics of this cohort prior to the pandemic, these observations raise concerns that a larger population of not yet infected susceptible individuals may lead to increased number of infections including RSV-associated hospitalization.

Treatment options for RSV infection are limited to symptomatic therapy. Only one drug, ribavirin, shows in vitro efficacy, however, has limited efficacy in patients and is therefore no longer recommended. A human monoclonal antibody (palivizumab) targeting the fusion protein has been used as prophylaxis to prevent infection of children at high risk of severe RSV infection (e.g., preterm neonates)[9]. However, this prophylaxis reduces hospitalization rates only by 55%, is costly and cannot be broadly applied. In addition, rapid development of resistance mutations in patients has been described[10]. Recently, nirsevimab

✉ e-mail: sibylle.haid@twincore.de; thomas.pietschmann@twincore.de

(trade name Beyfortus), a long-acting monoclonal antibody for the prevention of RSV infections in newborns and infants, was approved by several regulatory agencies around the world[11]. However, a need for new therapeutic options is still relevant.

Numerous antiviral strategies against RSV are in preclinical or clinical development[12–14]. These include immunoglobulins and small interfering RNAs as well as small molecules targeting viral proteins such as the glycoprotein G, the fusion protein F, nucleocapsid protein N, the transcriptional regulator protein M2-1 and the viral RNA-dependent RNA polymerase L[12–15].

Repurposing libraries accumulate licensed drugs or compounds in various stages of clinical development. Consequently, they are unique compound repositories, which offer potential opportunities for the rapid development of therapeutic applications. In case of SARS-CoV-2, screening of the ReFRAME library[16], the most comprehensive repurposing collection, has provided valuable orientation for development of coronavirus antivirals[17].

In this work, we screen the ReFRAME repurposing library and identify lonafarnib as RSV fusion protein inhibitor. We show that lonafarnib selects for resistance mutations within the RSV F protein, directly binds to RSV-F and thereby inhibits RSV membrane fusion and infection in vitro as well as in vivo.

## Results
### RSV reporter virus screen identifies lonafarnib as RSV inhibitor
To identify potential repurposing candidates, we interrogated the ReFRAME library encompassing 12,000 molecules using a recombinant RSV subtype A strain GFP reporter virus (Fig. 1A). We dosed

compounds at 5 μM and quantified infection efficiency based on GFP fluorescence at 48 h post inoculation. In parallel, we determined cell viability using an MTT assay. In total, we found 14 molecules, which met our primary hit criteria (RSV infection ≤ 16%; cell viability ≥ 80%) (Fig. 1B, dark blue dots). 37 molecules reduced RSV infection to background level, and exhibited variable degrees of cytotoxicity (Fig. 1B, dots at the very left). Speculating that at least some of these molecules may have an antiviral effect without cytotoxicity, if they are dosed at a lower concentration, we also included these compounds in our follow-up. Following the same rationale, we selected an additional 16 molecules based on a "floating" cell viability threshold (Fig. 1B, light blue dots). All of these latter molecules reduced RSV infection to <10%. Collectively, we reanalyzed 67 molecules by dose-response titration, and confirmed an antiviral activity separable from cytotoxicity for 21 molecules (Supplementary Figure S1). Besides well-known RSV inhibitors targeting the fusion protein (e.g. presatovir[18]) and the N protein (RSV-604[19]) (Fig. 1C and Supplementary Figure S1), we identified five heat shock protein 90 (HSP90) inhibitors, four Inosine-5-monophosphate dehydrogenase (IMPDH) inhibitors, a lipoprotein-associated phospholipase inhibitor (darapladib) and a farnesyl-S-transferase inhibitor (lonafarnib) (Supplementary Figure S2). To avoid library artifacts, we re-ordered the two latter compounds and two representatives for each the HSP90 and IMPDH inhibitors. We then confirmed their antiviral activity using an orthogonal infection assay based on a recombinant RSV subtype A strain Long luciferase reporter virus (Fig. 1A and C). Except for darapladib, we confirmed the antiviral activity of all these compounds. Among the confirmed candidates,

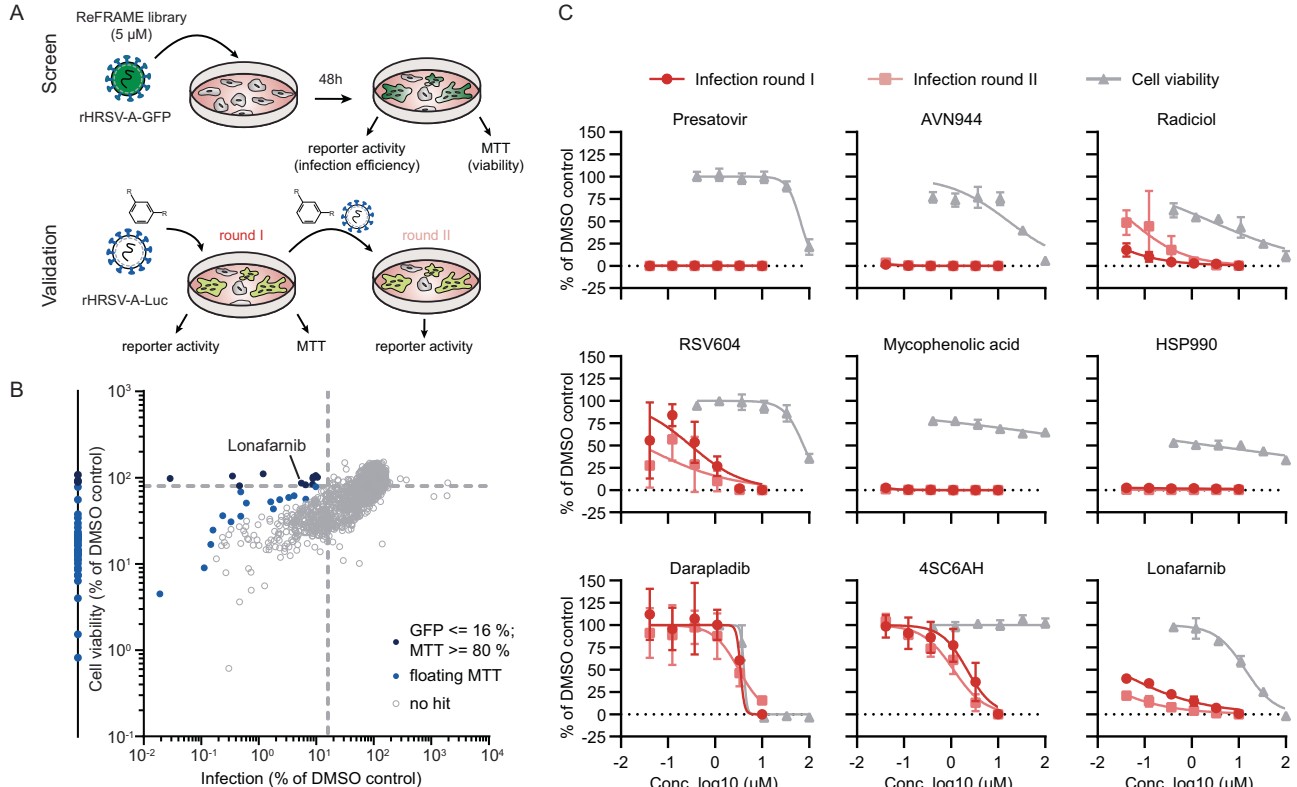

**Fig. 1 | Identification of drug repurposing candidates. A** Screening and validation procedure. **B** HEp-2 cells were infected with rHRSV-A-GFP[29] in presence of 5 μM compound. 48 hours later, infection and cell viability were quantified via GFP and MTT readouts. Dotted lines indicate primary hit criteria and dots represent means of two technical replicates. **C** HEp-2 cells were infected with HRSV-A-Luc[29] at MOI 0.01 and treated with the indicated compound concentrations. 24 hours later, supernatant was transferred onto new cells for a second round of infection.

Luminescence was quantified 24 hours post inoculation of both infection rounds. Cell viability was measured via MTT readout in treated, but uninfected cells. Mean ± SD of three independent experiments. Known RSV inhibitors (F protein: presatovir; N protein: RSV604, IMPDH inhibitors (AVN944, mycophenolic acid), HSP90 inhibitors (radiciol, HSP990). 4-Sulfocalix[6]arene Hydrate (4SC6AH, unknown target). Source data are provided as a Source Data file.

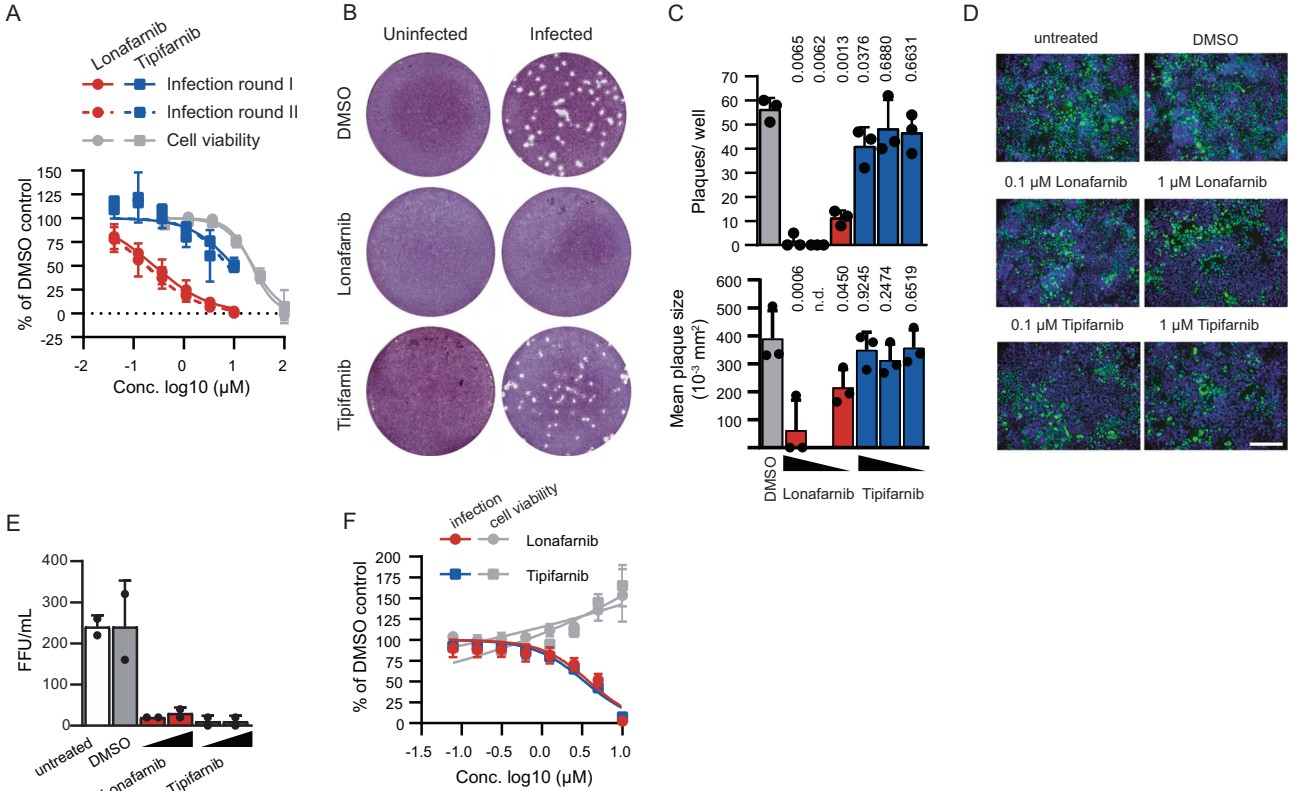

**Fig. 2 | Lonafarnib but not tipifarnib inhibits RSV infection.** **A** HEp-2 cells were infected with rHRSV-A-Luc[29] and treated with lonafarnib ($n = 4$) or tipifarnib ($n = 3$). Luciferase activity was measured and is expressed relative to the signal detected in DMSO treated infected cells. Mean ± SD of 3-4 independent experiments. MTT assay done $n = 3$ times. **B** HEp-2 cells were infected with a clinical isolate HRSV/A/DEU/H1/2013 in presence of 1 μM compound or DMSO control. At 6 dpi, monolayers were stained with crystal violet. Representative pictures of three independent experiments are shown. **C** Plaque number and mean plaque sizes were quantified using an ELISpot reader. Lonafarnib and tipifarnib were used at 1, 0.2 and 0.04 μM. Mean ± SD and individual results of three independent experiments. $P$ values from one-way repeated measures ANOVA with Dunnett´s multiple comparison

correction compared to DMSO are given. n.d., not determined due to absence of plaques. Huh7-hNTCP cells were transfected with HDV production constructs and treated with given compounds (0.1 and 1 μM). 10 days post transfection, translation of the HDV viral genome in the transfected cells was confirmed by immuno-fluorescence for the HD-Ag (**D**) (magnification: 10×). Production of HDV progeny was determined by inoculation of Huh7-hNTCP cells and staining of HD-Ag; scale bar: 200 μm. **E** Mean ± SD and individual results of two experiments are given. **F** Huh-7.5 F-luc cells were infected with hCoV-229E-Rluc[22] in presence of indicated compound concentrations. 48 hours later, infection and cell viability were measured by luciferase assays. Means ± SD and nonlin. fit from $n = 4$ independent experiments is given. Source data are provided as a Source Data file.

we selected lonafarnib, which is approved by EMA (https://www.ema.europa.eu/en/documents/product-information/zokinvy-epar-product-information_en.pdf) and FDA (https://www.accessdata.fda.gov/drugsatfda_docs/label/2020/213969s000lbl.pdf) for the treatment of Hutchinson-Gilford progeria syndrome and lamniopathies for further follow up. Lonafarnib is also in phase III clinical trials for the treatment of hepatitis delta virus infections (HDV) (https://clinicaltrials.gov/ct2/show/NCT03719313?term=lonafarnib&cond=HDV&draw=2&rank=4 and https://clinicaltrials.gov/ct2/show/NCT05229991?term=lonafarnib&cond=HDV&draw=2&rank=5).

**Lonafarnib but not tipifarnib inhibits RSV infection**
Chemically distinct farnesyltransferase inhibitors prevent lipidation of proteins such as Ras and progerin[20]. Therefore, they are being developed as candidates for anti-cancer therapy and for treatment of progeria. Moreover, farnesylation of the HDV capsid protein is critical for production of infectious virions, and lonafarnib is developed as an HDV antiviral[21]. Tipifarnib is also a farnesyltransferase inhibitor, which is developed as a cancer drug. Therefore, we examined if also tipifarnib inhibits RSV. However, only lonafarnib, but not tipifarnib inhibited infection by the recombinant RSV luciferase reporter virus (Fig. 2A) and a recent clinical RSV type A ON1 strain as determined by plaque reduction assays (Fig. 2B, C). Contrastingly, both compounds

equally inhibited release of infectious HDV progeny from Huh7-hNTCP cells and infection of Huh-7.5 F-luc cells by a hCoV-229E luciferase reporter virus[22], thus ruling out that tipifarnib was generally not active (Fig. 2D–F). To test if the antiviral effect of lonafarnib (and possibly tipifarnib) was RSV strain-dependent, we tested additional recent RSV subtype A and B isolates (Fig. 3). We sequenced early passages of these viruses, and conducted a phylogenetic analysis to assign these isolates to their cognate RSV subtype (Supplementary Figure S3). Additional metadata of the isolates and the accession numbers of the sequence data, deposited in the European Nucleotide Archive (ENA), can be found in the material methods section. This experiment revealed that lonafarnib is broadly active against recent clinical RSV subtype A and B strains, whereas tipifarnib was not antiviral for any one of the tested strains. The IC$_{50}$ of lonafarnib against these recent clinical strains ranged from 10-118 nM (Table 1). RSV-induced syncytia were readily discernable in DMSO treated infected cells, as was evidenced by detection of giant cellular aggregates characterized by clustered nuclei and shared cytosol (Fig. 3B, C). Interestingly, addition of lonafarnib suppressed syncytia formation (Fig. 3B, C). Given these findings and the divergent antiviral activity of lonafarnib/tipifarnib, we speculated that lonafarnib may inhibit RSV independent of blocking cellular farnesyltransferases and possibly through acting on a viral protein.

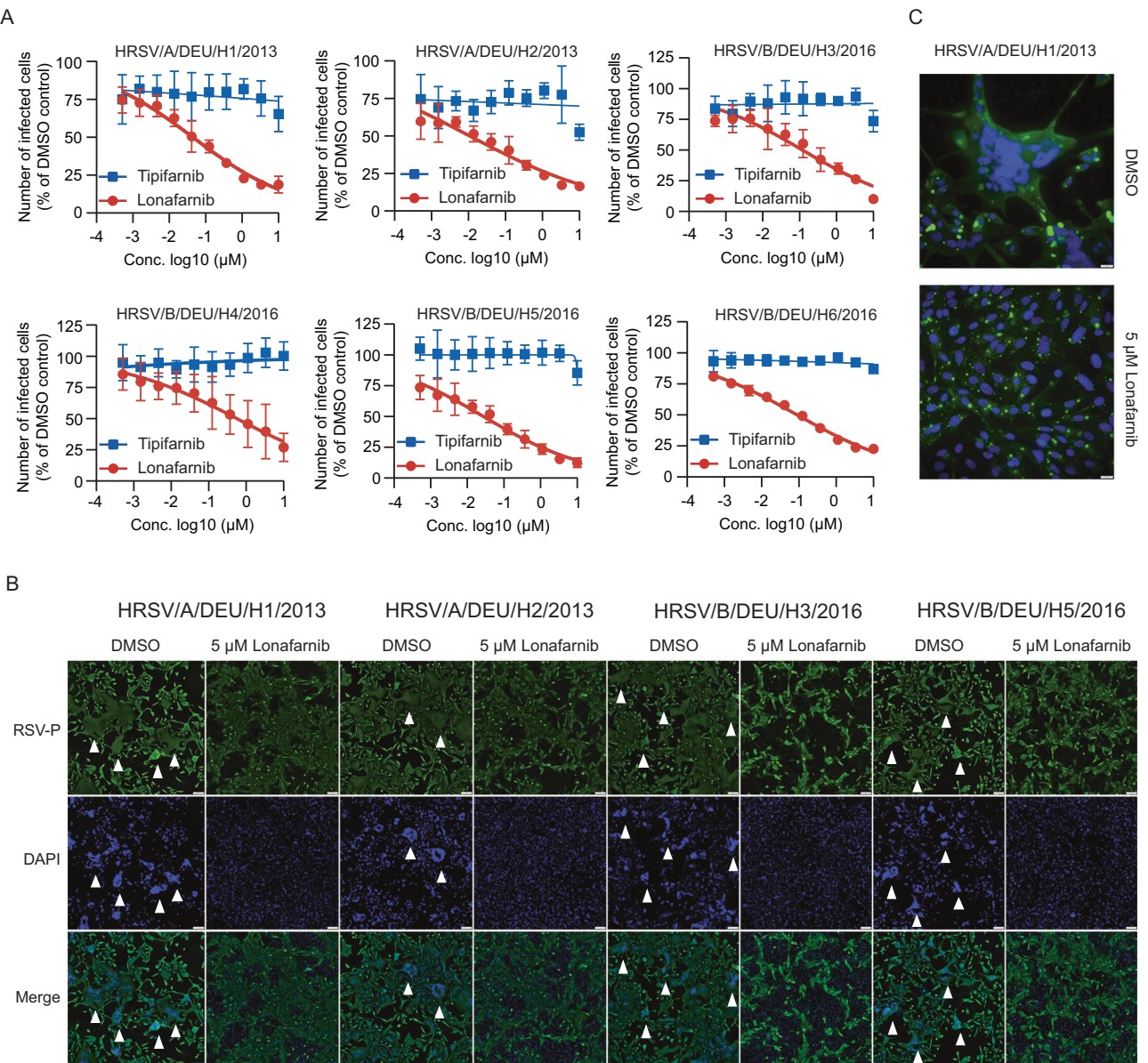

**Fig. 3 | Effect of lonafarnib on clinical RSV isolates. A** Dose-response curve of lonafarnib against 6 RSV isolates. HEp-2 cells were infected with RSV isolates with MOI of 1 (or 5 for HRSV/B/DEU/H6/2016) together with different concentrations of compounds. RSV infectivity was determined 24 h later by RSV-P protein staining and flow cytometry. Mean ± SD of three to ten biological replicates were given. ($n = 6$ for H1; $n = 3$ for H2, H5 and H6; $n = 4$ for H3; $n = 10$ for H4) (**B, C**) HEp-2 cells were inoculated with HRSV/A/DEU/H1/2013, HRSV/A/DEU/H2/2013, HRSV/B/DEU/H3/2016, or HRSV/B/DEU/H5/2016 4 h prior to treatment with 5 μM lonafarnib or

DMSO for 48 h. Cells were stained for RSV-P protein expression (green) and nuclear DNA (blue) (10× magnification) (**B**). Pictures from one of three independent experiments are given. Arrowheads highlight RSV induced syncytia. Scale bar: 100 μm. **C** Close-up pictures (40× magnification) of treated HEp-2 cells infected with HRSV/A/DEU/H1/2013. Scale bar: 20 μm. Representative pictures from one of two independent experiments are given. Source data are provided as a Source Data file.

**Table 1 | IC$_{50}$s and IC$_{90}$s of lonafarnib against RSV clinical isolates**

| | IC50 (μM) | 95% confidence interval | IC90 (μM) | 95% confidence interval |
|---|---|---|---|---|
| HRSV/A/DEU/H1/2013 | 0.04406 | 0.03329 to 0.05790 | 50.86 | 38.82 to 66.56 |
| HRSV/A/DEU/H2/2013 | 0.01041 | 0.005508 to 0.01813 | 180.6 | 106.6 to 303.7 |
| HRSV/B/DEU/H3/2016 | 0.1187 | 0.07228 to 0.1949 | 159.0 | 43.79 to 837.3 |
| HRSV/B/DEU/H4/2016 | 0.06392 | 0.04594 to 0.08871 | 255.8 | 185.4 to 352.8 |
| HRSV/B/DEU/H5/2016 | 0.02528 | 0.01595 to 0.03921 | 31.76 | 20.63 to 48.73 |
| HRSV/B/DEU/H6/2016 | 0.08874 | 0.07443 to 0.1058 | 168.8 | 142.0 to 200.5 |

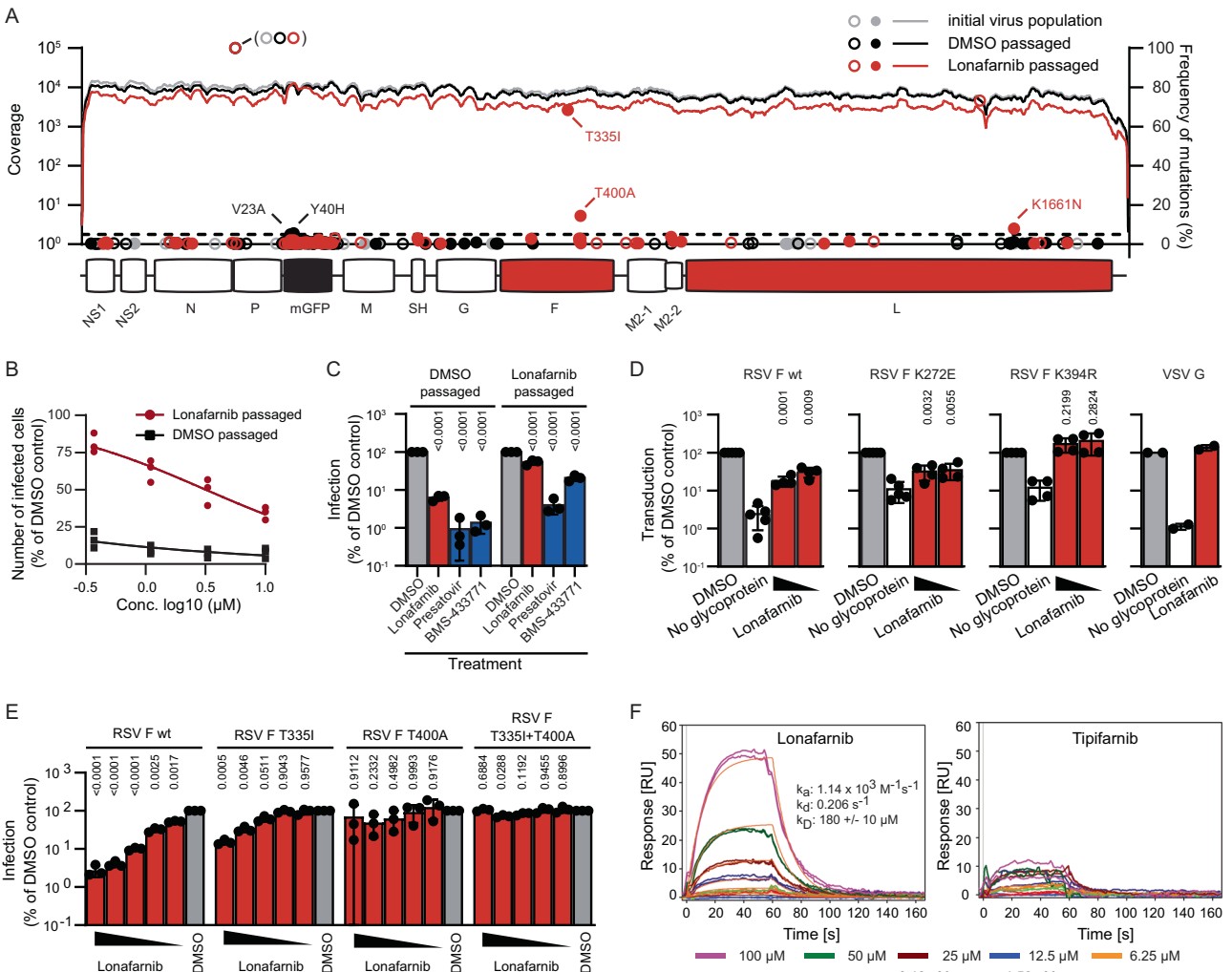

**Fig. 4 | Lonafarnib selects for RSV cross-resistance to entry inhibitors. A** rHRSV-A-GFP[29] virus populations were sequenced after 10 passages and compared to the initial sequence. Lines depict reads numbers across the genome; open circles non-coding, filled circles coding mutations. Amino acid exchanges with a frequency ≥ 5 % (dotted line) are labeled. **B** HEp-2 cells were infected with an MOI of 1 and infected cells quantified by flow cytometry. Symbols show results from three independent experiments. **C** HEp-2 cells were infected with the indicated virus population at an MOI of 1 and treated with 1 % DMSO, 10 μM lonafarnib, 0.1 μM presatovir or 10 μM BMS-433771. GFP-positive cells were quantified by flow cytometry. Mean ± SD and individual values of three independent experiments and *p* values are given. Statistical analysis was done by a 2way ANOVA with Sidák´s multiple comparison test in relation to DMSO control. **D** HEp-2 cells were transduced with lentiviral pseudoparticles in presence of DMSO or lonafarnib (5 μM and 0.5 μM). Mean ± SD and individual results of two to five independent experiments (*n* = 5 for DMSO- and

mock-treated RSVwt and K272E, *n* = 4 for lonafarnib-treated RSVwt and K272E and for all K394R, *n* = 2 for VSV-G pseudotypes) and *p* values. One-way repeated measures ANOVA with Dunnett´s multiple comparison test in relation to DMSO control. No statistics were calculated for the null-hypothesis (no glycoprotein). **E** HEp-2 cells were infected with lentiviral RSV F pseudotypes harboring the resistance mutations and lonafarnib (10 μM, 5 μM, 2 μM, 0.5 μM, 0.25 μM) or 1% DMSO. Mean ± SD and individual values of *n* = 3 independent experiments are given. *P* values from two-way ANOVA and Dunnett´s multiple comparison test in relation to DMSO control. **F** Surface plasmon resonance analyses of a prefusion RSV-F protein with lonafarnib (left) or tipifarnib (right) at concentrations of 1.56–100 μM (in duplicate) over an immobilized RSV subtype A pre-fusion F protein. In contrast to tipifarnib, lonafarnib shows significant and concentration-dependent binding responses to F protein. Binding kinetics and affinity were calculated by global fitting of the association and dissociation curves. Source data are provided as a Source Data file.

## Lonafarnib selects for resistance mutations in RSV F, inhibits cell to cell fusion and binds to the RSV F protein

To identify a potential viral target of lonafarnib, we passaged the RSV GFP reporter virus in presence of DMSO or increasing doses of lonafarnib and sequenced the resulting virus populations. Selectively, the lonafarnib-exposed virus population accumulated two coding mutations within the RSV fusion protein (T335I and T400A) (Fig. 4A) and developed phenotypic resistance to lonafarnib and two fusion protein inhibitors (presatovir and BMS-433771) (Fig. 4B–C). Next we used an RSV lentiviral pseudotype assay[23] to test if these changes affect susceptibility to lonafarnib. These RSV F-carrying pseudotypes were inhibited by lonafarnib and inhibition was abrogated by a known resistance mutation to RSV F protein inhibitors (K394R)[24,25] but not a

palivizumab resistance mutation (K272E)[26] (Fig. 4D). Strikingly, RSV pseudotypes with fusion proteins encoding the T335I or the T400A mutation or both displayed phenotypic resistance to lonafarnib, confirming that these changes confer resistance to lonafarnib (Fig. 4E). Finally, we used surface plasmon resonance and confirmed that lonafarnib, but not tipifarnib interacts with a recombinant RSV subtype A pre-fusion F protein (Fig. 4F). To further clarify the mode of action of lonafarnib against RSV, we conducted time of addition, RSV replicon and RSV F protein cell to cell membrane fusion assays (Fig. 5). Similar to ziresovir and palivizumab, lonafarnib was most effective when present only during virus inoculation (Fig. 5A). In contrast, ribavirin, a replication inhibitor, was most effective if it was applied 2 h after inoculation. In line with the assumption that lonafarnib inhibits cell

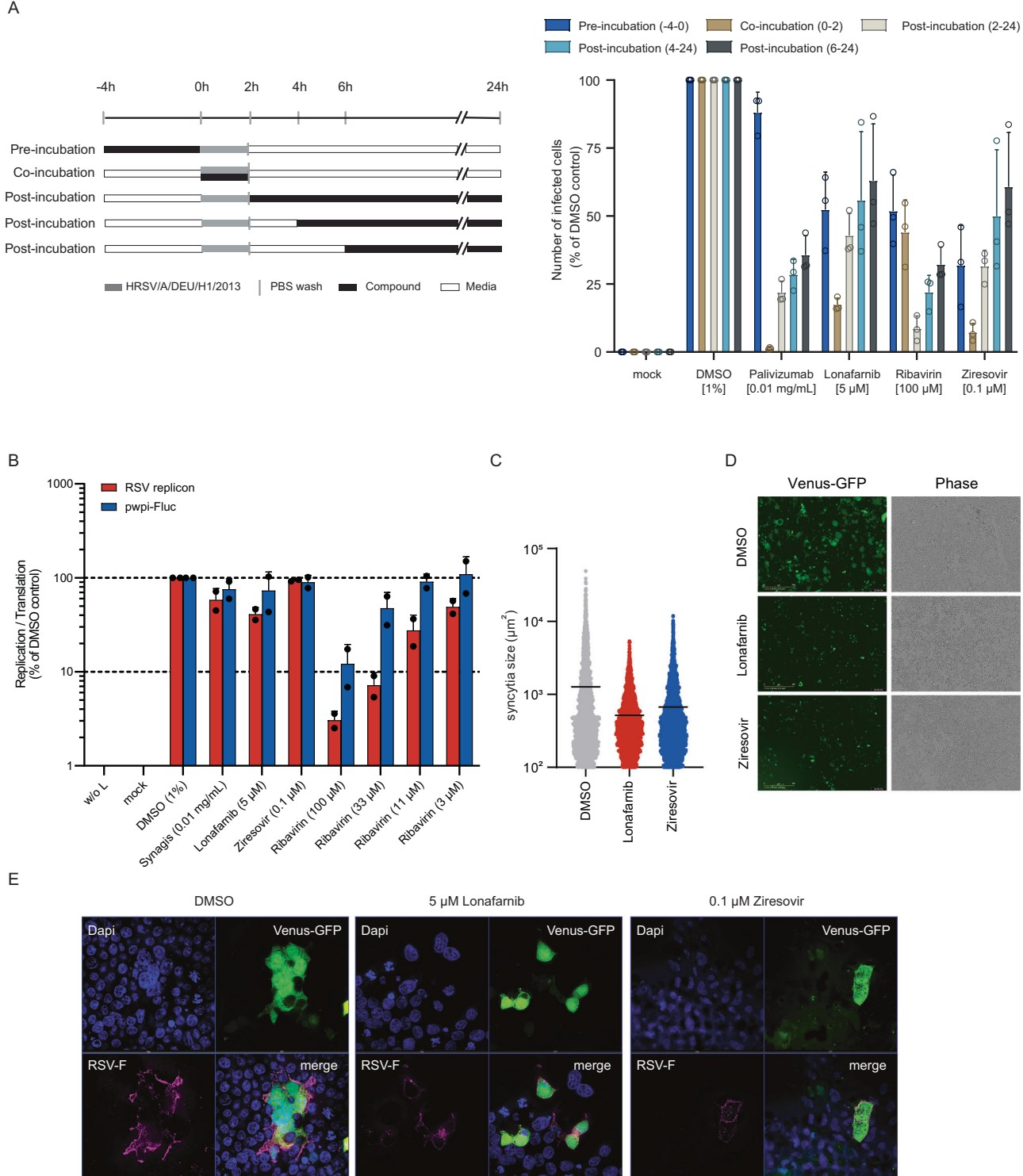

**Fig. 5 | Lonafarnib targets the fusion protein of HRSV. A** Time-of-addition assay of lonafarnib. HEp-2 cells were infected with HRSV/A/DEU/H1/2013 (MOI of 1) for 2 hours. Compounds were added as indicated. 24 hours later cells were harvested for intracellular staining of RSV-P and flow cytometry. Mean ± SD of three biological replicates were given. **B** Replicon assay. BSR-T7/5 cells transfected with either RSV replicon plasmids or system control plasmid pWPI-Fluc were treated with compounds containing media 4 hours post transfection, and luciferase activity was measured 3 days post compound treatment. Mean ± SD of two biological replicates were given. **C**–**E** Lonafarnib reduces cell-cell fusion induced by the RSV F protein. 293 T cells were transfected with a Venus-GFP and an RSV-F expression plasmid 6 h before treatment of cells with the solvent control DMSO (gray), 5 μM lonafarnib (red) or 0.1 μM ziresovir (blue). **C**, **D** 48 h post transfection, pictures were taken (10-fold magnification). All syncytia ( > 100 μm²) from four pictures per well of two wells per condition from a total of two independent experiments were analyzed using Fiji software. Means and symbols representing *n* = 11,803 syncytia examined over 2 independent experiments. **D** Representative pictures used to analyze (**C**). Scale bar: 400 μm. **E** 293 T cells were seeded on glass cover slips and co-transfected with a Venus-GFP and an RSV-F expression plasmid 6 h prior to treatment of cells with solvent control DMSO, 5 μM lonafarnib or 0.1 μM ziresovir. 72 h later cells were stained for RSV F protein (magenta) and DNA (blue) (100× magnification). Representative images from 2 independent experiments are given. Scale bar: 100 pixel equal to 10 μm. Source data are provided as a Source Data file.

entry, it did not significantly inhibit transcription and replication from an RSV minigenome (Fig. 5B). Finally, when we treated Venus-GFP and RSV F protein co-transfected cells with ziresovir or lonafarnib, both compounds significantly decreased the size of syncytia (Fig. 5C–E). Taken together these results indicated that lonafarnib inhibits RSV cell entry via binding to the fusion protein and by inhibition of membrane fusion; this inhibition is overcome by fusion protein resistance mutations.

### A co-crystal structure of RSV F in complex with lonafarnib identifies the interaction site

To better understand the interaction between lonafarnib and the fusion protein, we co-crystallized RSV F in complex with lonafarnib. Electron density for the compound was observed within the central cavity of prefusion F along the three-fold trimeric axis (Fig. 6 and Supplementary Figure S5), the crystallographic statistics are listed in supplementary table 1. Overall, lonafarnib binds to RSV F via the same hydrophobic pocket as previously described for other fusion inhibitors[27], with the planar aromatic groups of lonafarnib interacting with the aromatic side chains of F137, F140 and F488 located in the fusion peptide and the heptad repeats adjacent to the viral transmembrane region, respectively. These results unambiguously explain at an atomic level the interaction between lonafarnib and RSV F in a binding site that has been previously observed for other fusion inhibitors.

### Lonafarnib reduces virus load in a differentiated immortalized lung cell line

The only synthetic small molecule currently licensed for treatment of RSV infection is ribavirin. To better judge the potential of lonafarnib as a repurposing candidate, we first tested the potential of combining these two inhibitors. To this end, we systematically titrated combinations of these drugs and tested the antiviral activity using the RSV luciferase reporter virus. As is presented in Fig. 7A, we observed only minor inhibitory or slightly synergistic activity of combined compounds and only at selected doses (e.g. low dose of both compounds with slight inhibitory activity). In most combined doses, the antiviral effect was similar to the expected additive effect of these compounds.

Next, we tested if lonafarnib has a therapeutic effect, if applied after virus inoculation. To this end, we first inoculated A549 cells with HRSV-A-GFP. Twenty-four hours later we added DMSO, lonafarnib or ribavirin and followed the spread of the GFP-tagged reporter virus over time. Notably, treatment with lonafarnib restricted spread of the HRSV GFP virus down to ca. 30% of the DMSO treated control cells at 120 h post inoculation (Fig. 7B).

Finally, we examined if lonafarnib inhibits RSV infection also in a more natural model of RSV infection and cell entry. To this end, we took advantage of the immortalized human basal cell line BCi-NS1.1, which retains key characteristics of primary cells[28]. We differentiated these cells into a pseudostratified ciliated epithelium and infected them with the HRSV GFP reporter virus. As is shown in Fig. 7C, prophylactic treatment of these cells from both the apical and basolateral side dose-dependently inhibited RSV infection with a ca. 10- to 15-fold reduction of virus load in the infected cells and the culture fluid between 48 and 96 h after inoculation in presence of 5 μM lonafarnib. To extend these data, we tested if therapeutic application only from the basolateral side is sufficient to restrict infection and spread of RSV in this culture model. We infected the cells with a primary clinical RSV isolate (HRSV/A/DEU/H1/2013), added lonafarnib 24 h post virus inoculation (hpi) and measured RSV virus load in the wash fluid of the apical cell pole collected at 72 and 96 hpi. As is seen in Fig. 7D, therapeutic application of lonafarnib reduced virus load approximately to 50% of DMSO-treated control infections. Collectively, these data indicate that lonafarnib is therapeutically active in A549 and in differentiated BCi-NS1.1 cell culture models of RSV infection.

### Oral administration of lonafarnib reduces virus load in a mouse model of RSV infection

To provide proof of concept that lonafarnib can impair an RSV infection in vivo, we first conducted an orienting pharmacokinetic (PK) analysis after oral dosing of the molecule at 60 mg/kg. Using this regimen, we detected stable levels of lonafarnib in the epithelial lining fluid (ELF) of treated animals at levels greater than 1,000 μg/mL (equivalent to >1.56 μM) over a time frame of around 8 hours (Supplementary Figure S4A). An analysis of total plasma as well as ELF concentrations in relation to $IC_{50}$ and $IC_{90}$ values obtained for different RSV isolates suggests that this route of administration should allow accumulation of sufficient compound, in particular to ELF, to exert an antiviral activity (Supplementary Figure S4B). To test this, we treated groups of six animals with 60 mg/kg lonafarnib or the solvent control. Two hours later, we infected them with an RSV luciferase reporter virus[29] and monitored the course of infection by measuring bioluminescence on day 2 – 4 post-infection (dpi). We also measured the weight of the animals throughout this time frame; ultimately, we sacrificed the animals on 4 dpi and assessed the RSV copy numbers in the lung of animals. We repeated the experiment once more to confirm reproducibility, and collected lungs for histological analysis. As shown in Fig. 8, a 10-fold lower accumulation of lonafarnib in the lungs and bronchoalveolar lavage (BALF) of mice was detected in the first experiment compared to the second one (Figure 8A, B). However, these in vivo experiments showed a significantly reduced reporter virus signal in the lung and nose of treated animals throughout the time course (Fig. 8C–F). Furthermore, we observed a significant and dose-dependent decline of viral RNA in the lungs of treated animals at day 4 of the experiments (Fig. 8D). We also noted a trend that the lonafarnib-treated animals suffered less weight loss from the infection, more specifically in the first experiment (Fig. 8E). Histological analysis of mice lungs also revealed cellular infiltrates into the lungs of lonafarnib-treated animals (Fig. 8G). Taken together, these results provided a proof of concept that oral administration of lonafarnib exhibits an antiviral activity in vivo, whose efficacy correlated with drug levels in the lung tissue and BALF. However, when administered via the oral route, high doses of lonafarnib may have side effects caused by infiltrating cells.

## Discussion

In this study, we screened the ReFRAME compound library to identify drugs that might be repurposing candidates for treatment of RSV infections. Collectively, our screen of 12,000 molecules followed by an orthogonal confirmatory dose titration revealed 21 primary hit candidates (Supplementary Figure S1). Besides several directly acting antivirals to RSV F and N protein, which are in various stages of development, we identified five HSP90 and four IMPDH inhibitors. Among these hit candidates, the ReFRAME database lists hydrogen peroxide, and the IMPDH inhibitors mycophenolic acid sodium and mycophenolate mofetil as prescription drugs. While hydrogen peroxide is used topically as mild antiseptic on the skin or as a mouth gargle, the aforementioned IMPDH inhibitors are potent inhibitors of lymphoproliferation, which are used for prevention of allograft rejection. IMPDH is a key enzyme, which catalyzes a rate-limiting step in the de novo pathway of guanine nucleotides[30]. Given the dependence of many viruses on the availability of cellular nucleoside pools, several IMPDH inhibitors have emerged as potential broad antiviral agents[31–33]. Among these, Markland et al. previously reported that VX-497 is antiviral against RSV[33]. In line with this, VX-148, a derivative of VX-497, also emerged as hit in our screening. Collectively, these findings and our data emphasize the dependence of RSV on IMPDH.

HSP90 is a multiprotein complex that serves as a chaperone facilitating the folding of numerous client proteins[34]. Many viruses exhibit a dependence on HSP90 so that it has been proposed as broad-spectrum antiviral target[35–37]. In the case of RSV, HSP90 has been found in purified virus particles and HSP90 inhibitors were reported to

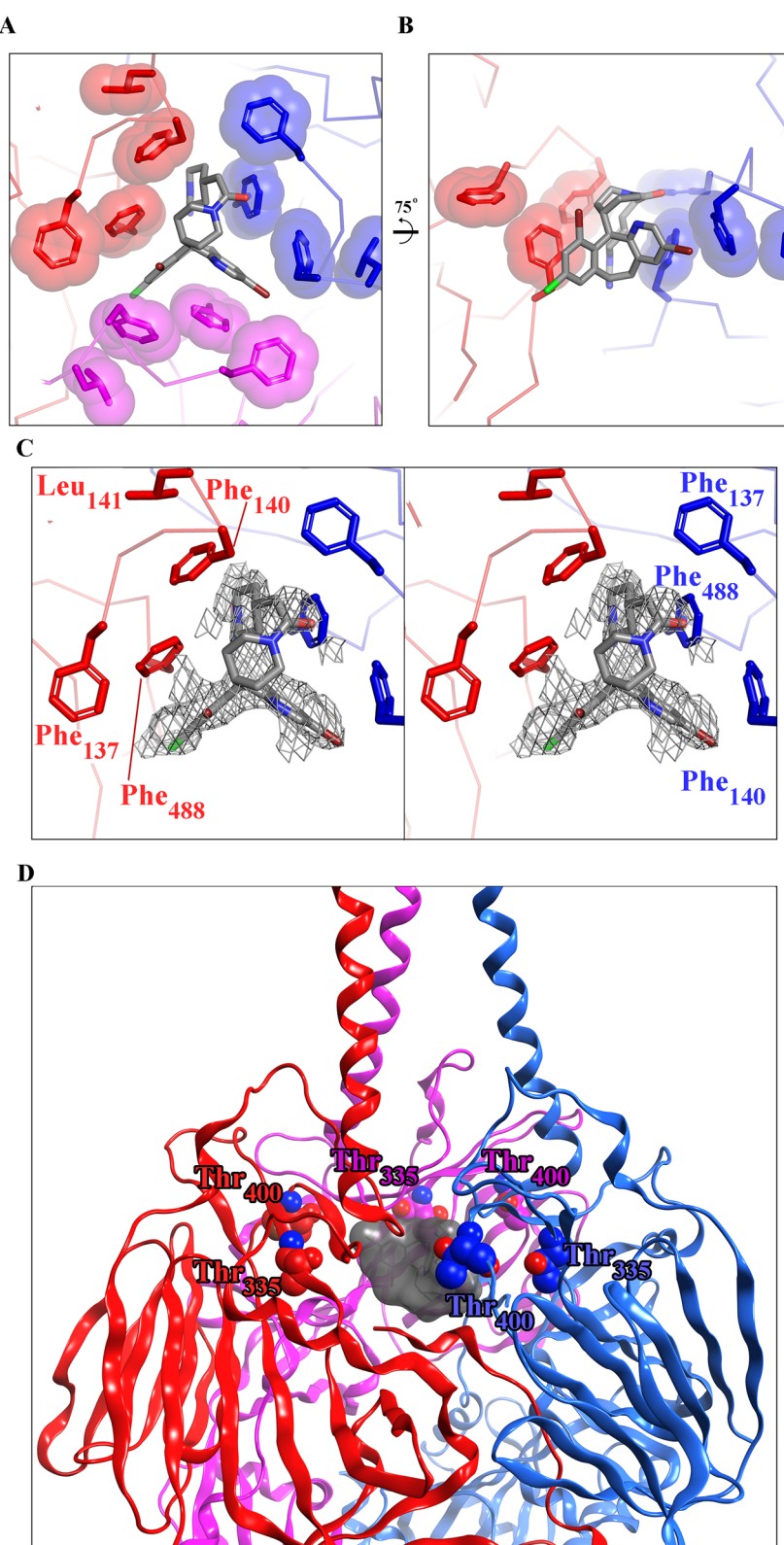

**Fig. 6 | Structure of RSV F in complex with lonafarnib.** Top (**A**) and side views (**B**) of lonafarnib bound to RSV F. Each F protomer is a colored differently (red, pink and blue), and hydrophobic side chains interacting with lonafarnib are shown with transparent molecular surfaces. In (**B**) one RSV F protomer is removed for clarity. One inhibitor molecule binds to each symmetry-related RSV-F protomer with an occupancy of 0.33, but for clarity only one inhibitor is shown as ball-and-stick model with carbon atoms colored in gray, nitrogen atoms in blue, oxygen atoms in red, and bromine atoms in dark red. **C** Stereo image of the top view including a polder map contoured at 3.0 sigma, around the ligand. **D** Zoomed-out view of the F-protein homotrimer in complex with lonafarnib (sticks and transparent gray surface) highlighting residues, which undergo resistance-conferring mutations upon lonafarnib exposure (T335 and T400; space filling-models).

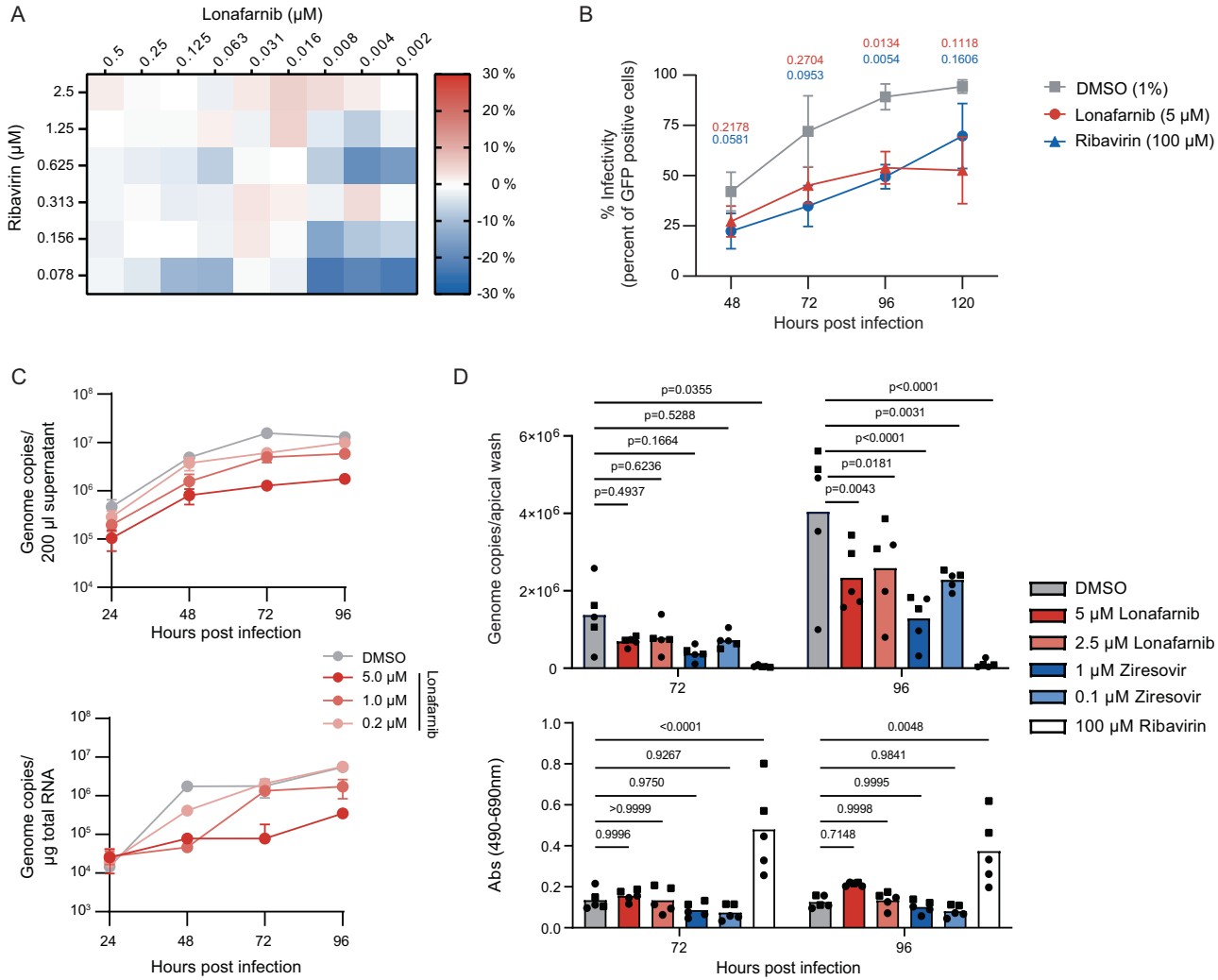

**Fig. 7 | Lonafarnib inhibits RSV infection in differentiated human lung cells.**
**A** HEp-2 cells were inoculated with rHRSV-A-Luc[29] (MOI of 0.01) together with given compounds. 24 h later, luminescence was quantified and the theoretical additive effects were calculated. The mean difference of the theoretical and measured combined effect of three independent experiments are shown. **B** A549 cells were infected with rHRSV-A-GFP[29] at an MOI of 0.01. 24 h later, cells were washed and supplemented with compound-containing media. RSV infection efficiency was determined at 48 h, 72 h, 96 h, 120 h post inoculation (i.e. 24 h to 96 h post addition of compounds) by measuring the number of infected cells using flow cytometry. Data were normalized to the highest number of infected cells as observed in the DMSO-treated specimen at 120 h post inoculation. Mean ± SD of three biological repeats were shown. Statistical analysis was done by 2way ANOVA and Dunnett´s multiple comparison test compared to DMSO data (p values). **C, D** Differentiated BCi-NS1.1 cells were grown as ALI cultures and treated (**C**) prophylactically or (**D**) therapeutically with the indicated compound concentrations. **C** One hour after

compound treatment from the basal side, cells were infected with rHRSV-A-GFP[29] from the apical side in presence of the indicated compound concentration for one hour. Apical compound treatment was repeated twice daily for one hour, basal treatment was repeated once daily for 24 hours. RSV RNA in supernatant (upper; mean ± SD of one to four technical replicates of one experiment; n = 4 for 24 h, n = 3 for 48 h, n = 2 for 72 h, n = 1 for 96 h) and cell lysates (lower graph; mean ± SD of one replicate measured in duplicates of one experiment) was quantified.
**D** Differentiated BCi-NS1.1 cells were inoculated with HRSV/A/DEU/H1/2013 (MOI 0.1) 24 h before treatment from the basolateral side. Apical washes were collected 72 h and 96 h later and a LDH toxicity analysis was performed. Viral genome copies were analyzed by qRT-PCR. Means of two independent experiments with 2 (square) or 3 (circles) transwells (i.e. technical replicates). Statistics were calculated in regard to DMSO treated cells using a 2-way ANOVA with Dunnett´s multiple comparison test. Source data are provided as a Source Data file.

decrease formation of virus particles[38]. Another study reported that HSP90 inhibition leads to degradation of the RSV polymerase protein L[39], suggesting that the folding and function of different RSV proteins depend on this chaperon. Extensive growth of RSV in presence of HSP90 inhibitors did not select for viral resistance, suggesting that RSV has a high barrier to develop resistance against these HSP90 inhibitors[39]. Overall, we found five known HSP90 inhibitors, which restricted RSV infection. Furthermore, glendanamycin and two of its analogs have already been observed to restrict RSV infection by other groups[38,39]. Further inhibitors of this ubiquitous protein have been tested in clinical trials and are being developed particular for cancer therapy[40,41]. Given the dependence of many different viruses on the

function of both IMPDH and HSP90 this previous work combined with our finding suggests that inhibitors of these cellular proteins merit attention for development of broad-spectrum antivirals.

In this study, we focused our attention on lonafarnib, which exhibited an $IC_{50}$ in HEp-2 cells against recent clinical RSV isolates ranging between 10-118 nM. So far, to our knowledge neither lonafarnib nor other farnesyltransferase inhibitors had been described as RSV antivirals. A previous study reported that the RSV fusion protein interacts with RhoA[42], a small GTPase that is prenylated; moreover, these authors reported that RhoA expression modulated RSV syncytium formation. Parallel to this, Gower et al. described that RSV infection triggers RhoA signaling and that inhibition of RhoA signaling

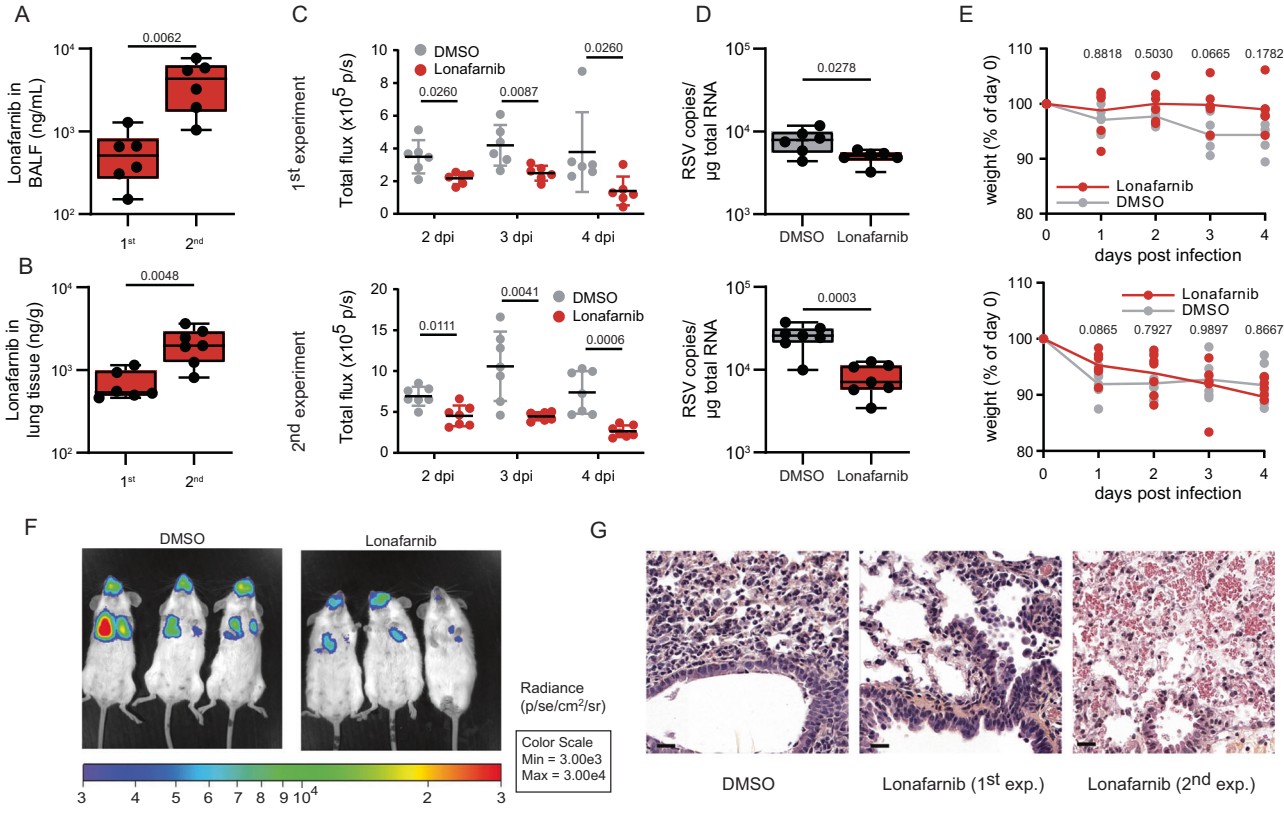

**Fig. 8 | Lonafarnib reduces RSV infection in mice. A–E** BALB/c mice were perorally treated with 60 mg/kg lonafarnib (1st experiment $N = 6$; 2nd experiment $N = 7$) or vehicle. Two hours later, mice were infected with a recombinant RSV luciferase reporter virus or mock infected. Drug treatment was repeated twice daily, subsequently. (**A/B**) Lonafarnib concentration in BALF (**A**) and the lung tissue (**B**) was quantified at 4 dpi. Note that for one mouse in the 2nd experiment there was no BALF available. (**A/B**) Boxes represent 25th to 75th percentiles including the median and whiskers go from minimum to maximum values. Dots represent individual mice. Two-tailed unpaired t-test. (**C**). Total bioluminescence in nose and lung for each animal. Each dot represents one animal. A Mann and Whitney (Houston) test was used. Data are presented as the mean ± standard error of the mean (SEM).

**D** Viral load in lung tissue at 4 dpi as quantified by qRT-PCR. Dots represent individual mice (1st experiment $N = 6$; 2nd experiment $N = 7$). Boxes represent 25th to 75th percentiles plus median and whiskers go from minimum to maximum values. Two-tailed unpaired t-test. **E** Mice bodyweight in percent of the respective starting weight. Dots represent individual mice. Two-way ANOVA followed by Sidak's multiple comparison test. For **C–E**, top row 1st and bottom row is 2nd experiment. **F** Exemplary pictures of bioluminescence for (**C**). **G** HES staining of mice lung treated with either DMSO or lonafarnib. Scale bar: 20 μm. Representative pictures of 2 independent experiments are given ($n = 2$). Source data are provided as a Source Data file.

prevents RSV induced cell to cell fusion[43,44]. Pastey et al. had reported that a RhoA-derived peptide inhibits RSV syncytia formation and infection both in vitro and in vivo[45]. Because of these findings and as the biological activity of RhoA depends on prenylation[46], we speculated that the antiviral activity of lonafarnib against RSV may be attributable to its capacity to inhibit cellular farnesyltransferases. While the results of this study do not exclude this possibility, the evidence provided here rather suggest a different mode of action. First, a chemically distinct farnesylation inhibitor (tipifarnib) did not affect RSV infection. Second, a typical resistance mutation against RSV F inhibitors conferred resistance to lonafarnib in a lentiviral RSV pseudotype infection assay. Congruently, selection of RSV in presence of lonafarnib caused accumulation of mutations within the RSV fusion protein in areas previously implicated in development of fusion protein inhibitor resistance. Third, lonafarnib treatment reduced syncytia formation in infected cells and in F protein over-expressing cells. Finally, we observed direct binding of lonafarnib (but not tipifarnib) to recombinant RSV F protein and we resolved its binding site within the RSV F protein trimer. Although these results do not rule out that inhibition of farnesyltransferases contributes to the antiviral effect of lonafarnib, these data provide strong support for the conclusion that lonafarnib binds the RSV F protein and inhibits F protein-dependent membrane fusion and in turn infection.

To explore the potential of lonafarnib as a candidate for drug repurposing, we confirmed the antiviral activity with recent RSV subtype A and B clinical isolates. In addition, we ruled out that combination treatment with the only currently licensed small molecule inhibitor of RSV (ribavirin) has overt adverse effects. At the level of the in vitro HEp-2 cell culture assay, we did not observe a major interference of these drugs with each other. Using A549 cells and the more authentic differentiated air liquid interface BCi-NS1.1 tissue culture model, we confirmed that RSV infection of a pseudostratified epithelium is inhibited by lonafarnib, both in a prophylactic and a therapeutic treatment regimen. However, it should be noted that RSV infection of primary human lung cells ex vivo depends on C-X3-C motif chemokine receptor 1 (CX3CR1)[47]. Likewise, mouse and cotton rat infection by RSV in vivo depend on CX3CR1[47,48]. Given that it is currently not known if the BCi-NS1.1 cell model recapitulates RSV-CX3CR1-dependence, caution is warranted when extrapolating these in vitro data to the complex viral receptor dependence in human lung cells. Finally, we provided proof of concept that lonafarnib is antiviral in a mouse model of RSV infection. Please note that only female animals were used in these in vivo experiments so that we cannot rule out a gender specific effect on these results. The two experiments that we conducted suggest that the antiviral activity correlates with lonafarnib exposure in lung tissue and BALF. In particular, our PK study results show that doses of

lonafarnib in the epithelial lining fluid (ELF) are above the $IC_{90}$ values of several RSV strains tested in this study (Supplementary Figure S4B). Moreover, total plasma concentrations after oral administration were above the determined $IC_{50}$ values of several strains for several hours suggesting that total plasma levels above $IC_{50}$ could be a good surrogate for efficacy.

A number of RSV F-targeting fusion inhibitors have been described and therapeutic efficacy of this class of inhibitors was established in various in vitro and in vivo models[12]. Some of these molecules, including presatovir (GS-5806), rilematovir (JNJ-53718678), sisunatovir (RV521), enzaplatovir (BTA-C585), and ziresovir (AK0529) have advanced to human clinical trials. However, lack of therapeutic benefit in adult hematopoietic stem cell transplant recipients and a low barrier to resistance has led to the discontinuation of the development of presatovir[49–52]. Moreover, Janssen recently made the strategic decision to terminate the DAISY trial, a phase 3 study exploring the efficacy of rilematovir in children and infants with acute RSV infection (NCT04583280). In contrast, the development of sisunatovir continues, and a phase 3 study of ziresovir in hospitalized infants was recently completed (NCT04231968). Given the moderate number of clinical stage candidates and high attrition rates in antiviral drug development, new drug candidates are urgently needed. Our data show that lonafarnib potently inhibits RSV infection in vitro. However, lonafarnib´s efficacy is lower compared with above mentioned clinical stage inhibitors (lonafarnib $IC_{50}$ range against recent clinical strains from 10 to 118 nM compared to: rilematovir $EC_{50}$ = 0.5 nM; sisunatovir $EC_{50}$ = 1.3 nM, ziresovir $EC_{50}$ = 5 nM[52]). Furthermore, lonafarnib also inhibits farnesyltransferases and may therefore have unwanted side effects, particularly when administered orally and at high doses. This potential concern is stressed by our result showing that a 10-fold increased deposition of lonafarnib in the BALF correlated with enhanced antiviral activity but also side effects as plasma levels are close to the in vitro determined $CC_{50}$ of lonafarnib. According to a PK/PD study on lonafarnib treated chronic HDV patients[53], serum concentration of lonafarnib in patients administered with 100-200 mg lonafarnib twice daily can reach 0.4-1.68 $\mu$M at 4-6 hours post intake (Cmax 256 ng/ml at 4 h, Cmax 1073 ng/ml at 6 h), which covers all of the $IC_{50}$ values of the RSV clinical isolates in this study (Table 1). This indicates that if given orally at early stages of infection, lonafarnib is likely to be effective in limiting RSV infection and propagation. Finally, the plasma concentrations in mice reached values slightly higher than 1 $\mu$M, which might be the cause for side effects. It is possible that alternative routes of lonafarnib administration improve the efficacy/side effect ratio. To explore this, the testing of alternative application routes and formulations could be useful. For instance, it is possible that inhalation of lonafarnib deposits high compound levels directly to the apical side of lung cells, where infection and cell to cell spread occurs. This route of administration may improve efficacy with a tolerable degree of side effects, because tissue-wide access to the host target, that is likely at least in part responsible for unwanted effects, may be reduced compared to the oral administration route.

## Methods
### Media, cells, viruses and compounds
HEp-2 cells (ATCC CCL-23), A549 cells (ATCC CCL-185), 293 T (ATCC CCL3216) cells and Huh-7.5 F-luc cells were cultured in Advanced MEM (HEp-2), F12K NutMix (A549) or DMEM media (293 T, Huh-7.5 F-luc) respectively supplemented with 10% heat-inactivated FCS (Capricorn Scientific), 1% NEAA (Gibco), 2 mM L-glutamine (Gibco) as well as 100 U/ml Penicillin and 100 U/ml Streptomycin (Gibco) at 37 °C and 5% $CO_2$ in a humified incubator. Huh-7.5 cells[54] were a kind gift from Charles M. Rice (Rockefeller University, USA). These cells were engineered to express a firefly luciferase gene by lentiviral gene transfer[55]. BCi-NS1.1 cells[28] were cultivated and differentiated as described elsewhere[56]. Recombinant reporter viruses rHRSV-A-Luc[29], rHRSV-A-GFP[29] and hCoV-

229E-Rluc[22] were described elsewhere. Palivizumab was obtained from AbbVie Ltd (North Chicago, IL), ribavirin from Sigma-Aldrich and lonafarnib from BLDpharm. BMS-433771 was a kind gift from Richard Karl Plemper (Georgia State University, Atlanta, USA). BSR-T7/5 cells were a kind gift from Karl-Klaus Conzelmann (Ludwig-Maximilians-University Munich, Germany).

### High throughput screening using ReFRAME library
The HTS was performed using a Biomek FX Automation Workstation. 3 × $10^3$ HEp-2 cells were seeded in 60 $\mu$l media in black non-transparent 384-well plate the day prior to infection. The next day, a recombinant HRSV-A reporter virus was mixed with compounds and 20 $\mu$l of the mixture was added to the cells resulting in the indicated final compound concentration and an MOI of 0.5. Uninfected DMSO-treated as well as infected DMSO-treated cells and cells treated with 2 $\mu$g/ml palivizumab served as controls. 48 h post infection, supernatant was transferred onto new, naïve HEp-2 cells for second round infection and the cells were washed with PBS and fluorescence was quantified using a BioTek Cytation 3 cell imaging multi-mode reader at 485 nm and 528 nm. After fluorescence quantification as marker for RSV replication, cell viability was analyzed in each well by addition of 50 $\mu$l 1 mg/ml MTT in growth media for 90 min at 37 °C prior to lysis of cells in 50 $\mu$l isopropanol and absorbance measurements at 595 nm and 630 nm. Cell viability was normalized to uninfected, DMSO treated cells. The second round of infection was also stopped at 48 h post inoculation and fluorescence quantification was performed as above. No cell viability was analyzed for the second-round infection.

### Recombinant protein production
Recombinant RSV F protein derived from the A2 strain in a prefusion conformation stabilized by the structure-based design of disulfide (DS) and cavity-filling (Cav1) mutations (McLellan et al. [57]) was produced in Drosophila melanogaster S2 cells. A gene encoding the stabilized glycoprotein was cloned into a modified Drosophila S2 expression vector described previously and transfection was performed as reported earlier[58]. For large-scale production, cells were induced with 4 mM $CdCl_2$ at a density of approximately 4×$10^6$ cells/ml to 8×$10^6$ cells/ml for 4 days, pelleted, and the soluble trimeric F ectodomain was purified by affinity chromatography from the supernatant using a StrepTactin Superflow column followed by size exclusion chromatography using a Superose 6 column equilibrated in 20 mM HEPES, 150 mM NaCl. Pure protein was concentrated to approximately 5 mg/ml.

### Crystallization and structure determination
Crystals of prefusion-stabilized RSV F were grown at 291.5 K using the hanging-drop vapor-diffusion method in drops containing 1 $\mu$l protein mixed with 0.l $\mu$l lonafarnib (10 mM in 100% DMSO) and 1 $\mu$l reservoir solution containing 20% PEG6000, 400 mM $CaCl_2$, and 100 mM Tris-HCl pH 8. Diffraction quality rhombohedral crystals appeared after 3 weeks and were flash-frozen in mother liquor containing 30% (v/v) ethylene glycol. Data collection was carried out at beamline Proxima-1 of the Synchrotron Soleil. Data were processed, scaled, and reduced with XDS[59], Pointless[60] and programs from the CCP4 suite[61]. A single-wavelength anomalous dispersion (SAD) dataset was collected from a single RSV F crystal at the K edge of bromine (0.91983 Å) using low-dose, high-redundancy (5× 360 degrees) fine-sliced collection strategy[62] using five crystal orientations at different chi angles by means of a high-precision multi-axis SmarGon goniometer. The structure was determined by the molecular replacement method using Phaser[63] and PDB 5EA5 as search model. Subsequent model building was performed using Coot[64], and refinement was done using Auto-Buster 2.10.4[65] with repeated validation using MolProbity[66]. The final Ramachandran statistics for favored, allowed and outliers are 96%, 4% and 0%, respectively. Clear electron density was observed for residues 26 – 69, 73 – 99 and 137 – 518 with disordered loops 61 – 69, 211 – 216,

210 – 253, 220 – 332, 385 – 394 and 448-491. As the placement of an asymmetric inhibitor on a 3-fold crystallographic symmetry axis was difficult, we combined the phases of the fully refined model with the anomalous differences derived from the two bromine atoms in the lonafarnib molecules to calculate an anomalous difference map. The major anomalous difference map peak revealed the position of the first bromine atom, and the fact that only a single anomalous difference map peak was observed together with the geometry of the ligand indicated that the second bromine atom was located on an almost symmetry-related position rotated around the 3-fold axis. Based on this placement of the two bromine atoms the intact inhibitor was placed. To calculate a polder map, the ligand occupancy was set to 0, a single refinement run performed and the final polder map created using Phenix[67]. Figures were prepared with Pymol software (http://www.pymol.org/).

### Ethics statement

The in vivo studies in mice were carried out in accordance with the INRAE guidelines, which are compliant with the European animal welfare regulation. The animal studies were conducted in accordance with the recommendations of the European Community (Directive 86/609/EEC, 24 November 1986, EU Directive 2010/63/EU). All animal procedures were performed in strict accordance with the German regulations of the Society for Laboratory Animal Science (GV-SOLAS) and the European Health Law of the Federation of Laboratory Animal Science Associations (FELASA). The protocols were approved by the Animal Care and Use Committee at "Centre de Recherche de Jouy-en-Josas" (COMETHEA) under relevant institutional authorization ("Ministère de l'éducation nationale, de l'enseignement supérieur et de la recherche"), authorization number 201803211701483v2 (APA-FIS#14660) or the ethical board of the Niedersächsisches Landesamt für Verbraucherschutz und Lebensmittelsicherheit, Oldenburg, Germany. All experimental procedures were performed in a biosafety level 2 facility.

### RSV in vivo experiment

Female BALB/c mice were purchased from the Centre d'Elevage R. Janvier (Le Genest Saint-Isle, France). All mice were group housed 2-5 in polypropylene cages in a standard temperature- and humidity-controlled biosafety laboratory 2 animal facility with a 12 h light-dark rhythm, unlimited access to food and water and enrichments (nests). Cages, food, enrichment and water were sterilized before use. Mice at 8 weeks of age ($n = 6$ or 7 per group) were treated by gavage with 60 mg/kg of lonafarnib resuspended in 20% DMSO, 40% PEG400, 40% Hydroxy-propyl-beta-cyclodextrin-solution (Sigma). Two hours later, mice were anesthetized with a mixture of ketamine and xylazine (1 and 0.2 mg per mouse, respectively) and treated IN (intranasal) with 60 µl of rHRSV-Luc ($10^5$ p.f.u.). A second treatment was administrated 10 hours later, and mice were treated twice at 1, 2, and 3 day. Luminescence measurement was performed at 2, 3, and 4 dpi.

### In vivo luminescence measurements

Mice were anesthetized at 2, 3, and 4 dpi and bioluminescence was measured 5 min following instillation of 50 µl D-luciferin (30 mg/ml, Perking Elmer). Living Image software (version 4.0, Caliper Life Sciences) was used to measure the luciferase activity. Bioluminescence signals were acquired with an exposure time of 1 min. Digital false-color photon emission images of mice were generated and show the average radiance (p/s/cm²/sr). Photons were counted within three different regions of interest corresponding to the nose, the lungs and the whole airway area. Signals are expressed as total normalized flux (p/s). All data were analyzed using the GraphPad Prism software version 6.07. The statistical significance for all in vivo bioluminescence experiments were measured using the Mann and Whitney (Houston) test. All data are presented as the mean ± standard error of the mean

(SEM) and the $p$ values. The number of individuals and repeated experiments are stated in each figure legend.

### Histological analysis

The mice were sacrificed at 4 dpi, the chest cavity was opened, and the lungs were perfused intratracheally with 4% paraformaldehyde (PFA) in PBS. The lungs were then removed and immersed in 4% PFA for 12 h before transfer in 70% ethanol. The lungs were embedded in paraffin, and 5 µm sections were cut, stained with hematoxylin-eosin-saffron (HES), and evaluated microscopically. Qualitative histological changes were described and, when applicable, were scored semi quantitatively using a three-point scale ranging from 0 to 2 (0, none; 1, mild; 2, marked), focusing on histological characterization of the lesion (interstitial pneumonia, respiratory epithelial cell apoptosis, and hyperplasia) and inflammation.

### Lonafarnib PK analysis

Lonafarnib was dissolved in 20% DMSO, 40% PEG400, 40% hydroxypropyl-beta-cyclodextrin in water (20/80 (w/v)). Mice were administered lonafarnib at 60 mg/kg p.o. using an intragastric gavage. At the time points 0.5, 1, 2, 4, 8, and 24 h post administration, mice were euthanized to collect blood from the heart and spontaneous urine as well as to perform bronchoalveolar lavage (BALF) and to remove lungs aseptically. Whole blood was collected into Eppendorf tubes coated with 0.5 M EDTA and immediately spun down at 13,000 rpm for 10 min at 4 °C. Then, plasma was transferred into a new Eppendorf tube and lungs were homogenized using a Polytron tissue homogenizer. BALF, lung and plasma samples were stored at −80 °C until analysis.

### Bioanalysis of PK and in vivo efficacy samples

All PK and in vivo efficacy samples were analyzed via HPLC-MS/MS using an Agilent 1290 Infinity II HPLC system and coupled to an AB Sciex QTrap6500plus mass spectrometer. First, a calibration curve was prepared by spiking different concentrations of lonafarnib into mouse plasma (pooled, from CD-1 mice), urine, homogenized lung or isotonic sodium chloride solution (the latter one served as matrix for BALF samples). Caffeine was used as an internal standard. In addition, quality control samples (QCs) were prepared for lonafarnib in the respective matrices. The following extraction procedure was used: 7.5 µl of a plasma sample (calibration samples, QCs or PK samples) was extracted with 25 µl of a 1:1 mixture of methanol and acetonitrile containing 12.5 ng/ml of caffeine as internal standard for 5 min at 2,000 rpm on an Eppendorf MixMate® vortex mixer. For urine samples, 15 µl of a urine sample (calibration samples, QCs, or PK samples) was extracted with 25 µl of a 1:1 mixture of methanol and acetonitrile containing 12.5 ng/ml of caffeine as internal standard for 5 min at 2,000 rpm on an Eppendorf MixMate® vortex mixer. 100 µl of methanol was added to 50 µl of each BALF sample (calibration samples, QCs, PK or efficacy samples) and dried for 2 hours in an Eppendorf concentrator using vacuum and 25 °C. Then, 15 µl of water and 35 µl of a 1:1 mixture of acetonitrile and methanol containing 12.5 ng/ml caffeine as internal standard was added. Samples were shaken for 10 min on an Eppendorf MixMate® vortex mixer. 50 µl of a lung sample adjusted to a concentration of 50 mg/ml with isotonic sodium chloride solution (calibration samples, QCs, PK or efficacy samples) was extracted with 50 µl of a 1:1 mixture of acetonitrile and methanol containing 12.5 ng/ml caffeine as internal standard for 5 min on an Eppendorf MixMate® vortex mixer. Then samples (plasma, urine, BALF or lung) were spun down at 13,000 rpm for 10 min. Supernatants were transferred to standard HPLC-glass vials. HPLC conditions were as follows: column: Agilent Zorbax Eclipse Plus C18, 50×2.1 mm, 1.8 µm; temperature: 30 °C; injection volume: 5 µl; flow rate: 700 µl/min; solvent A: water + 0.1% formic acid; solvent B: acetonitrile + 0.1% formic acid; gradient: 99% A at 0 min and until 0.1 min, 99% − 20% A from 0.1 min to 3.0 min, 20% − 0% A from 3.0 min to

**Table 2 | Mass spectrometric conditions for analytes**

| ID | Q1 Mass [Da] | Q3 Mass [Da] | time [msec] | CE [volts] | CXP [volts] | DP [volts] |
|---|---|---|---|---|---|---|
| lonafarnib | 638.982 | 596.000 | 50.0 | 25.0 | 28.0 | 1.0 |
|  |  | 621.900 | 50.0 | 41.0 | 28.0 | 1.0 |
|  |  | 470.000 | 50.0 | 43.0 | 22.0 | 1.0 |
| caffeine | 195.024 | 138.000 | 30.0 | 25.0 | 14.0 | 130.0 |
|  |  | 110.000 | 30.0 | 31.0 | 18.0 | 130.0 |
| Urea | 60.915 | 43.800 | 30 | 17.0 | 16.0 | 56.0 |
|  | 60.915 | 43.100 | 30 | 53.0 | 12.0 | 56.0 |
|  | 60.915 | 29.100 | 30 | 111.0 | 6.0 | 56.0 |

3.3 min, 0% A until 4.0 min, 0% − 99% A from 4.0 min to 4.5 min, 99% A from 4.5 min to 4.7 min. Mass spectrometric conditions were as follows: Scan type: MRM, positive mode; Q1 and Q3 masses for caffeine and lonafarnib can be found in Table 2; peak areas of each sample and of the corresponding internal standard were analyzed using Multi-Quant 3.0 software (AB Sciex). Peaks of PK and in vivo efficacy samples were quantified using the calibration curve. The accuracy of the calibration curve was determined using QCs independently prepared on different days. ELF concentrations were calculated using the following formulas:

$$V_{ELF} = V_{BALF} \times \frac{Urea_{BALF}}{Urea_{Plasma}} \tag{1}$$

$$c_{ELF} = c_{BALF} \times \frac{V_{BALF}}{V_{ELF}} \tag{2}$$

**Statistical analysis**

Statistical analysis was done as outlined in each specific material and methods section. Error bars and replicates are defined in the respective figure legends.

**Reporting summary**

Further information on research design is available in the Nature Portfolio Reporting Summary linked to this article.

## Data availability

Reagents can be requested by formal application. Requests should be directed to the corresponding authors. The accession numbers for the sequencing raw reads in the NCBI Sequence Read Archive (SRA) are SRR22746624, SRR22746625, and SRR22746626. The atomic coordinates and structure factors for the crystal structure was deposited in the Protein Data Bank (https://www.rcsb.org) under the accession number 8PHI. The sequence of RSV clinical isolates used for this study have been deposited in the European Nucleotide Archive (ENA) at EMBL-EBI under accession number PRJEB63686. Source data are provided as a Source Data file. Source data are provided with this paper.

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

## Acknowledgements

We thank the animal facilities of Infectiology of fishes and rodents (IERP, INRAE, doi: 10.15454/1.5572427140471238E12), and the Emerg'in platform for access to IVIS200 that was financed by the Region Ile De France (SESAME). This work has benefited from the facilities and expertise of @BRIDGe (Université Paris-Saclay, INRAE, AgroParisTech, GABI, Jouy-en-Josas, France). We thank Andrea Ahlers, Kimberley Vivien Sander and Janine Schreiber for excellent technical assistance. We acknowledge

SOLEIL for provision of synchrotron radiation facilities, and we would like to thank Andrew Thompson for assistance in using beamline PROXIMA-1. We are grateful to Richard Karl Plemper (Georgia State University, Atlanta, USA) for BMS-433771 and to Charles M. Rice (Rockefeller University, New York, USA) for provision of Huh-7.5 cells. Twincore is a joint venture between the Medical School Hannover (MHH) and the Helmholtz Centre for Infection Research (HZI).

A.K.H.H., T.K., C.L., T.P., G.H. and T.F.S. are funded by the Deutsche Forschungsgemeinschaft (DFG, German Research Foundation) under the Germany's Excellence Strategy—EXC 2155 "RESIST"—Project ID 390874280. T.P. and G.H. are also funded by the INDIRA project (11-76251-99-6/19 (ZN3437). T.P. and A.K.H.H. also receive funds from the Helmholtz International Lab for anti-infectives. K.R. receives funding from the German Center for Infection Research (DZIF, TTU 09.719). T.P., A.K.H.H., T.K. and G.H. are funded by the Volkswagen Foundation (OPTIS, Project ID 9B811). The funders had no role in study design, data collection and interpretation, or the decision to submit the work for publication.

## Author contributions

Conceptualization: SMS, SH, TP, MG, KR, Methodology: SMS, MKR, MUQ, SH, XZ, AC, APG, JR, MG, TL, RLG, FH, KR, WAME, JH, NF, MARW, AKHH, ME, CL, EH, TK, TP, Validation: SMS, SH, MKR, XZ, AC, APG, AM, JR, MG, TL, RLG, FH, KR, WAME, JH, NF, ME, AKHH, CL, EH, TK, TP, Formal analysis: SMS, XZ, MUQ, SH, MKR, AC, APG, AM, MG, TL, RLG, FH, KR, WAME, JH, CL, NF, AKHH, EH, TK, TP, Investigation: SMS, MKR, MUQ, SH, XZ, AC, APG, CG, JR, MG, TL, KR, WAME, JH, EH, Resources: JFE, MARW, TFS, AKC, KJ, KM, MW, GH, Data curation: SMS, MKR, MUQ, SH, XZ, AC, APG, AM, MG, TL, AKC, KR, WAME, JH, EH, Writing original draft: SMS, SH, TP, Writing review and editing: SMS, SH, XZ, AC, APG, AM, MG, JFE, KR, WAME, AKHH, NF, MARW, TK, TFS, TP Visualization: SMS, SH, XZ, AC, AM, CL, ME, TL, WAME, Supervision: SMS, SH, NF, AKHH, TK, TFS, TP, Project administration: SMS, SH, TP, Funding acquisition: KR, AKHH, TK, TFS, TP, All authors read and approved the manuscript.

## Competing interests

SMS, SH, JR, TFS and TP disclose that they are authoring a patent application "LONAFARNIB FOR USE IN THE TREATMENT OF VIRAL INFECTIONS" describing lonafarnib as antiviral compound that inhibits RSV infection (21152993.8-1132). The authors declare no competing interests and no restrictions on the publication of data.

## Additional information

Svenja M. Sake[1], Xiaoyu Zhang[1], Manoj Kumar Rajak[2,3], Melanie Urbanek-Quaing [1], Arnaud Carpentier [1], Antonia P. Gunesch[1], Christina Grethe[1], Alina Matthaei[1], Jessica Rückert[2], Marie Galloux[4], Thibaut Larcher [5], Ronan Le Goffic [4], Fortune Hontonnou[4], Arnab K. Chatterjee[6], Kristen Johnson [6], Kaycie Morwood[6], Katharina Rox [7,8], Walid A. M. Elgaher [9,10,11], Jiabin Huang [12], Martin Wetzke[13,14], Gesine Hansen[11,13,14], Nicole Fischer[12], Jean-Francois Eléouët[4], Marie-Anne Rameix-Welti [15], Anna K. H. Hirsch [9,10,11,16], Elisabeth Herold[3], Martin Empting [8,9,10,11], Chris Lauber [1,11], Thomas F. Schulz [2,8,11], Thomas Krey [2,3,17,18], Sibylle Haid [1] ✉ & Thomas Pietschmann [1,8,11,16] ✉

[1]Institute for Experimental Virology, TWINCORE, Centre for Experimental and Clinical Infection Research, Hannover, Germany. [2]Institute of Virology, Hannover Medical School, Hannover, Germany. [3]Center of Structural and Cell Biology in Medicine, Institute of Biochemistry, University of Luebeck, Luebeck, Germany. [4]Université Paris-Saclay, INRAE, UVSQ, VIM, Jouy-en-Josas, France. [5]INRAE Oniris, PAnTher, APEX, Oniris, Nantes, France. [6]Calibr, Scripps Research, La Jolla, CA, USA. [7]Department of Chemical Biology, Helmholtz Center of Infection Research, Braunschweig, Germany. [8]German Centre for Infection Research, Partner site Braunschweig-Hannover, Braunschweig, Germany. [9]Helmholtz Institute for Pharmaceutical Research Saarland (HIPS)—HZI, Saarbrücken, Germany. [10]Department of Pharmacy, Saarland University, Saarbrücken, Germany. [11]Cluster of Excellence RESIST (EXC 2155), Hannover Medical School, Hannover, Germany. [12]Insitute for Medical Microbiology, Virology and Hygiene, University Medical Center Hamburg-Eppendorf, Hamburg, Germany. [13]Department for Pediatric Pneumology, Allergology and Neonatology, Hannover Medical School, Hannover, Germany. [14]German Center for Lung Research, Partner Site Hannover, BREATH, Hannover, Germany. [15]Université Paris-Saclay, Université de Versailles St. Quentin; UMR 1173 (2I), INSERM; Assistance Publique des Hôpitaux de Paris, Hôpital Ambroise Paré, Laboratoire de Microbiologie, DMU15, Versailles, France. [16]Helmholtz International Lab for Anti-infectives, HZI, Braunschweig, Germany. [17]Centre for Structural Systems Biology (CSSB), Hamburg, Germany. [18]German Center for Infection Research, Partner Site Hamburg-Luebeck-Borstel-Riems, Luebeck, Germany. ✉e-mail: sibylle.haid@twincore.de; thomas.pietschmann@twincore.de

