## [Peer Review File · Nature Communications]

Drug repurposing screen identifies lonafarnib as respiratory syncytial virus fusion protein inhibitorReviewer #1 (Remarks to the Author):

This is a well written manuscript that reports on the identification of lonafarnib, a known inhibitor of farnesyl transferase with an IC₅₀ of 1.9nM, as a fusion protein inhibitor against RSV. The manuscript report on the drug repurposing library containing 12,000 small molecules that was screened with the identification of 21 candidates against RSV. Lonafarnib was selected for further development because it is licensed as a farnesyl transferase inhibitor for the treatment of a rare disorder, Hutchinson-Gilford progeria syndrome, and is being studied for treatment of a variety of cancers and hepatitis delta virus. In a series of in vitro and in vivo studies lonafarnib was shown to prevent infection with RSV by acting on the F protein. The development of resistance was also shown after multiple passages in the presence of lonafarnib. The major resistance was identified in the T400A mutation. Studies in mice demonstrated variable drug bioavailability in BALF and lung tissue (Fig 4D) with variable antiviral effect (Fig 4G) and potential toxicity (Fig 4J).

There are several major limitations that impact the interpretation of lonafarnib as an antiviral candidate for RSV. First, data were not presented to demonstrate the effectiveness of lonafarnib for the treatment of an RSV infection. All known antiviral drug candidates are being developed for the treatment of an infection and not for the prevention. Studies demonstrating the treatment effect are crucial if the drug is to be developed further. Secondly, antiviral effects using a panel of contemporary RSV/A/Ontario and RSV/B/Buenos Aires strains should be considered and an IC₅₀ and IC₉₀ should be established. It is unclear how effective lonafarnib is against wild type RSV/A and RSV/B strains compared to GFP or luciferase reporter viruses. Lastly, variable drug bioavailability and toxicity are concerning and potentially may be overcome with formulation changes and or route of exposure.

Lonafarnib at concentration of 1 μM, 0.2μM and 0.04μM were tested against RSV/A/ON and RSV/B with plaque forming unit as the end point. Please indicate the IC₅₀ and IC₉₀ by the plaque assay for these two viruses. Also indicate in the discussion section, if these levels can be reached by oral administration of lonafarnib. Also please provide some metadata on these two isolates such as date of isolation and to what genotype RSV/B belongs.

It is unclear why the combined apical and basolateral routes of exposure were used in the ALI cultures. Ideally, one route (basolateral or apical) should be used to generate the data. For these studies the basolateral route would be more appropriate and simulate the mouse study with the gavage or oral route of exposure. Also, the BCI-NS1.1 ALI platform is an ideal platform to study antiviral drugs to RSV because of the long duration of viral shedding. Exposure to the drug after the infection is established is the preferred method for demonstrating an antiviral effect compared to treatment prior to the infection. Again, the study design used was not optimal for showing an antiviral treatment effect.

The RSV luciferase reporter virus used for the mouse studies is an excellent addition to study antiviral drugs against RSV. However, it is also important to demonstrate if the luciferase activity corresponds to the antiviral effect on infectious RSV in both the lungs and nose. This information was not provided. Likewise, the use of a wild type strain should have also been used to validate the findings of the RSV luciferase reporter virus.

Reviewer #2 (Remarks to the Author):

Review Nature Communications manuscript NCOMMS-22-52752

The manuscript from Svenja M et al. described a RSV reporter virus screening system to identify RSV inhibitors from the ReFRAME library encompassing 12,000 molecules. 21 drugs were identified as RSV inhibitor. One of the hits, lonafarnib, was chosen for further analysis. The authors analyzed the effects of lonafarnib on RSV replication in cell model and mouse model, highlighting a positive role of lonafarnib in anti-RSV replication. Additionally, the authors proposed that lonafarnib inhibits RSV by acting on the RSV fusion protein rather than by inhibition of cellular farnesyltransferases. However, the logic of the article is not clear enough, the phenotypic and mechanistic study of lonafarnib on RSV are not deep enough, so the conclusions are not convincing. Overall, the present studies are preliminary in nature, the enthusiasm for this study is limited, and there are several important issues that need to be addressed.

Major comment:

(1) In terms of mechanisms, the authors directly examined the effect of lonafarnib on RSV fusion

protein (F protein) after excluding that lonafarnib did not affect RSV replication by inhibiting farnesyltransferase activity, and finally concluded that lonafarnib inhibited RSV infection by inhibiting F protein. This conclusion is less convincing. The authors should design experiments to verify which stage in the RSV life cycle is inhibited by lonafarnib (entry, uncoating, replication packaging, budding or release). For example, a time-of-addition assay that lonafarnib is added at different time points before and after RSV infection can be employed.

(2) The authors found that lonafarnib can directly bind to the fusion protein, but did not investigate the effect of lonafarnib on the function of the fusion protein. RSV fusion protein plays an important role in the viral entry and membrane fusion, so the authors can evaluate the effect of lonafarnib on virus-cell fusion and cell-cell fusion. For instance, the authors can detect the effect of lonafarnib on the syncytia formation. The known inhibitor of the RSV fusion protein, presatovir, could also be used as a positive control to perfect the experiments. Furthermore, before the RSV infection, the virions are first adsorbed to the cell surface and then initiate the virus-cell fusion. Thus, the authors can employ viral attachment experiments to assess whether lonafarnib blocks RSV attachment.

(3) In the Figure 2, to assess the effect of lonafarnib on RSV infection, in addition to the Plaque assay, the authors should detect other indicators that can reflect RSV replication, such as the expression level of RSV P protein or N protein can be detected by Western blot, the mRNA level of RSV can be detected by the RT-qPCR or Northern blot.

(4) In the Figure 3E, The authors used the SPR assay to verify the interaction between fusion protein and lonafarnib. In order to make this conclusion more convincing, we believe that the interaction between lonafarnib and fusion protein mutant (K394R) should be supplemented here as a control and other experiments (such as ITC, BLI or CETSA) should be added to further verify the interaction. Moreover, if possible, the authors need to judge the specific binding sites between lonafarnib and fusion protein by the crystal structure analysis.

(5) In the Figure 4, the authors used RSV luciferase reporter virus or RSV GFP reporter virus to infect the immortalized lung cell lines and mouse model. We believe that there are differences between genetically modified viruses and natural RSV strains. Thus, the author should complement the natural RSV-infected cells and animal experiments. Moreover, in animal studies, the authors should increase the experimental regimen of dosing after infection as a control and add Ribavirin as a positive control to compare the antiviral effect with lonafarnib.

Minor comment:

(1) Why the IC50 of lonafarnib differed 5-fold between the first round and second round of infection.

(2) In the Figure 2A-F, the authors should focus on the phenotypic studies of RSV inhibition by lonafarnib, while the results of the effect of Tipifarnib on RSV should be placed in the Supplementary Data.

(3) In the Figure 2G, the results of lonafarnib and Tipifarnib affecting HDV replication should be placed in the Supplementary Data.

(4) The discussion section of the article should be rewritten to focus on the development of RSV fusion protein inhibitors and studies related to lonafarnib.

Reviewer #3 (Remarks to the Author):

This manuscript describes the screening of a drug repurposing library for inhibitors of respiratory syncytial virus (RSV) infections. The authors identified a number of candidates which had been previously identified as inhibitors of various host cell proteins. The authors focused on one, lonafarnib, a farnesyltransferase inhibitor, as a potential inhibitor of RSV infections. The authors described inhibition of RSV infections in tissue culture cells and the toxicity of the drug to tissue culture cells. They showed that lonafarnib bound to purified soluble prefusion RSV F protein and that the inhibitor selected drug resistant mutations after 10 passages of RSV in tissue culture. Experiments in mice were presented to show efficacy of the inhibitor in vivo. Many of the experiments presented are moderately well done, particularly experiments done in tissue culture, and the results raise very interesting questions for future studies.

There are, however issues with the study that decrease enthusiasm for the development of this

inhibitor for RSV.

1. The authors did not show inhibition of infections when the inhibitor was added after the virus. That is, they did not show time of addition of the inhibitor on inhibition relative to addition of virus in tissue culture.
2. In a related issue they only tested in mice the potential for prophylactic effect of inhibitor treatment but not its therapeutic potential.
3. In mouse experiments, the authors used BalbC mice which are only semi permissive to RSV infection. Testing in more permissive animals, such as cotton rats, would be a more rigorous test of the potential of the inhibitor. Furthermore, in tests of efficacy in mice, the authors added the inhibitor multiple times during the course of the experiment. Was this necessary to see an effect?
4. The legend or text does not describe the protocol used for results shown in Figure 4, panel J. It appears that there is considerable lung pathology in drug treated animals, a result inconsistent with inhibition of infection in the animals.
5. The authors did not show direct evidence that the drug blocked membrane fusion as would be expected if it inhibited cell entry, as the authors imply.
6. The conclusion that inhibition by lonafarnib is not due to its activity on farnesyltransferase is circumstantial and is premature without further experiments.

Other issues:

1. There is a lack of statistical analysis of some of the results. Examples are: Figure 2, panels H, C, F; Figure 3, panels C and D; Figure 4 panel C.
2. Some of the experiments are not well described and are difficult to follow. The figure legends or text should more completely describe protocols of experiments.
3. The authors only used female animals.
4. Figure 2, panel D bottom left, why is the fluorescence increasing with concentration of inhibitor?

Reviewer comments and point-by-point replies

Reviewer #1 (Remarks to the Author):

This is a well written manuscript that reports on the identification of lonafarnib, a known inhibitor of farnesyl transferase with an IC₅₀ of 1.9nM, as a fusion protein inhibitor against RSV. The manuscript report on the drug repurposing library containing 12,000 small molecules that was screened with the identification of 21 candidates against RSV. Lonafarnib was selected for further development because it is licensed as a farnesyl transferase inhibitor for the treatment of a rare disorder, Hutchinson-Gilford progeria syndrome, and is being studied for treatment of a variety of cancers and hepatitis delta virus. In a series of in vitro and in vivo studies lonafarnib was shown to prevent infection with RSV by acting on the F protein. The development of resistance was also shown after multiple passages in the presence of lonafarnib. The major resistance was identified in the T400A mutation. Studies in mice demonstrated variable drug bioavailability in BALF and lung tissue (Fig 4D) with variable antiviral effect (Fig 4G) and potential toxicity (Fig 4J).

There are several major limitations that impact the interpretation of lonafarnib as an antiviral candidate for RSV. First, data were not presented to demonstrate the effectiveness of lonafarnib for the treatment of an RSV infection. All known antiviral drug candidates are being developed for the treatment of an infection and not for the prevention. Studies demonstrating the treatment effect are crucial if the drug is to be developed further.

Response: We thank this referee for the diligent evaluation of our study and for raising this important point. We acknowledge this limitation of our previous manuscript and therefore conducted additional experiments in order to confirm the effectiveness of lonafarnib for the treatment of an RSV infection. First, we inoculated A549 cells with RSV-A-GFP at an MOI of 0.01. Twenty-four hours later, we washed the cells and added fresh medium containing antiviral compounds. Subsequently at 48h, 72h, 96h, 120h post inoculation (i.e. 24h up to 96h upon addition of compounds), we determined RSV infection efficiency by measuring the number of infected cells using flow cytometry. Data were normalized relative to the highest number of infected cells observed in DMSO control treated cells at 120h post inoculation. At this terminal timepoint and with therapeutic treatment of 5 μ M lonafarnib, RSV infected only ca. one third of the cell numbers infected in the DMSO-treated cells. This infection rate is somewhat lower compared to treatment with Ribavirin (100 μ M), which reduced infected cell number to ca. 50% as compared to the DMSO control (compare novel Figure 7B).

Second, we differentiated BCI-NS1.1 cells and cultured them in air liquid interface (ALI) configuration. Subsequently, we inoculated the cells with a recent clinical RSV-A strain (HRSV/A/DEU/H1/2013) at an MOI of 0.1 24h prior to treatment from the basolateral side. Apical washes were collected 72h and 96h post inoculation and a LDH toxicity analysis was performed from the basolateral media (lower graph). Viral genome copies were analyzed by qRT-PCR analysis. Under these conditions, 5 and 2.5 μ M lonafarnib reduced RSV genome equivalents ca. by 50% in the apical washes of treated BCI-NS1.1 ALI cultures. These novel data were included as novel Figure 7D. Collectively, these results provide evidence, that therapeutic application of lonafarnib after virus inoculation inhibits RSV infection and spread in A549 and differentiated BCI-NS1.1 cells.

Secondly, antiviral effects using a panel of contemporary RSV/A/Ontario and RSV/B/Buenos Aires strains should be considered and an IC50 and IC90 should be established. It is unclear how effective lonafarnib is against wild type RSV/A and RSV/B strains compared to GFP or luciferase reporter viruses.

Response: We acknowledge this important suggestion of the referee and conducted additional experiments using different recent RSV A and RSV B clinical isolates. These new results are included as novel Figure 3A. As requested by the referee, we show the IC50 and IC90 values of Lonafarnib against these recent clinical RSV isolates in the novel table 1. These new results confirm that lonafarnib is antiviral against recent clinical RSV A and B strains.

Lastly, variable drug bioavailability and toxicity are concerning and potentially may be overcome with formulation changes and or route of exposure.

Lonafarnib at concentration of 1 μ M, 0.2 μ M and 0.04 μ M were tested against RSV/A/ON and RSV/B with plaque forming unit as the end point. Please indicate the IC50 and IC90 by the plaque assay for these two viruses. Also indicate in the discussion section, if these levels can be reached by oral administration of lonafarnib. Also please provide some metadata on these two isolates such as date of isolation and to what genotype RSV/B belongs.

Response: Thank you for the comment. To provide a more comprehensive view of lonafarnib's activity against RSV, we performed additional experiments using recent primary clinical RSV isolates. We sequenced these isolates (the sequence of RSV clinical isolates used for this study have been deposited in the European Nucleotide Archive (ENA) at EMBL-EBI under accession number PRJEB63686 (<https://www.ebi.ac.uk/ena/browser/view/PRJEB63686>) and confirmed that one of the RSV A strains belongs to genotype ON1 and the other one to the genotype NA1. All tested B strains cluster among BA11 genotype.

To precisely quantify the IC50 and IC90 values of lonafarnib against these viruses, we performed infections in presence of increasing doses of lonafarnib and quantified the number of infected cells by intracellular FACS staining of the RSV P protein. The dose response curves of these titrations are given in novel Figure 3A. The IC50 and IC90 values are summarized in the new table 1. To better illustrate the effect of lonafarnib on cellular morphology after RSV infection, we performed fluorescence staining of the RSV-P protein at 48h post infection and compound treatment (Fig. 3B). This analysis revealed decreased syncytia formation and provides first evidence that lonafarnib may inhibit fusion.

According to a PK/PD study on lonafarnib treating chronic HDV patients (DOI: 10.1002/hep4.1043), serum concentration of lonafarnib in patients administered with 100-200 mg lonafarnib twice daily can reach 0.4-1.68 μ M at 4-6 hours post intake (Cmax 256 ng/ml at 4h, Cmax 1073 ng/ml at 6h), which covers most of the IC50s, but not the IC90s of lonafarnib against RSV clinical isolates (Table 1), indicating that if given orally at early stages of infection, lonafarnib is likely to be effective in limiting RSV infection and propagation. Moreover, our PK data from the animal studies (novel supplementary figure S4) show that plasma concentrations cover the IC50 values of the strains assessed *in vitro*. Additionally, ELF concentrations show that they are above the IC90 values of the RSV strains assessed *in vitro*. Furthermore, as ELF concentrations follow plasma kinetics, this suggests that plasma concentrations above IC50 might be a good surrogate for efficacy. Finally, the plasma

concentrations are reaching values slightly higher than 1 μ M which might be the cause for side effects.

It is unclear why the combined apical and basolateral routes of exposure were used in the ALI cultures. Ideally, one route (basolateral or apical) should be used to generate the data. For these studies the basolateral route would be more appropriate and simulate the mouse study with the gavage or oral route of exposure. Also, the BCI-NS1.1 ALI platform is an ideal platform to study antiviral drugs to RSV because of the long duration of viral shedding. Exposure to the drug after the infection is established is the preferred method for demonstrating an antiviral effect compared to treatment prior to the infection. Again, the study design used was not optimal for showing an antiviral treatment effect.

Response: We acknowledge this helpful and constructive advice. In addition to the prophylactic treatment, we now performed experiments in a therapeutic setting and using differentiated BCI-NS1.1 cells. First, we infected the cells with RSV (MOI 0.1). Twenty-four hours later we treated the infected cells with compounds only from the basolateral side (novel Fig. 7D). We could show that at 96h post infection, lonafarnib was able to significantly reduce RSV propagation and release to the apical side compared to the DMSO control. In addition, we were also able to show in A549 cells that lonafarnib added after RSV infection could reduce the level of RSV propagation and spread. Together, these results provide further evidence that lonafarnib treatment inhibits RSV infection.

The RSV luciferase reporter virus used for the mouse studies is an excellent addition to study antiviral drugs against RSV. However, it is also important to demonstrate if the luciferase activity corresponds to the antiviral effect on infectious RSV in both the lungs and nose. This information was not provided. Likewise, the use of a wild type strain should have also been used to validate the findings of the RSV luciferase reporter virus.

Response: Thank you for the comment. In the paper by Rameix-Welti et al., 2017, we have shown that there is a perfect correlation between luciferase signal and viral mRNA by qRT-PCR. This was confirmed in Descamps et al., 2021 (<https://doi.org/10.1128/JVI.00912-21>). The RSV-Luc system is now largely validated and widely used and confirmation by RT-PCR has proven to be no longer necessary like in Jacque et al., 2021 (doi: 10.3389/fimmu.2021.683902); Risso-Balester et al., 2021 (<https://doi.org/10.1038/s41586-021-03703-z>), Palsson et al., 2020 (doi: 10.3389/fimmu.2020.580547), Galloux et al., 2020 (<https://doi.org/10.1128/AAC.00717-20>), Laubreton et al., 2020 (doi:10.3390/v12080822), Blockus et al., 2020 (<https://doi.org/10.1016/j.antiviral.2020.104774>), Bryche et al., 2019 (<https://doi.org/10.1111/jnc.14936>), Gaillard et al., 2017 (<https://doi.org/10.1128/AAC.02241-16>), Cagno et al., 2017 (DOI: 10.1038/NMAT5053), Hervé et al., 2016 (<http://dx.doi.org/10.1016/j.jconrel.2016.10.003>). The recombinant virus has the same sequence as the Long strain except the insertion of a luciferase gene between P and M genes (Rameix-Welti et al., 2017).

Reviewer #2 (Remarks to the Author):

Review Nature Communications manuscript NCOMMS-22-52752

The manuscript from Svenja M et al. described a RSV reporter virus screening system to identify RSV inhibitors from the ReFRAME library encompassing 12,000 molecules. 21 drugs were identified as RSV inhibitor. One of the hits, lonafarnib, was chosen for further analysis. The authors analyzed the effects of lonafarnib on RSV replication in cell model and mouse model, highlighting a positive role of lonafarnib in anti-RSV replication. Additionally, the authors proposed that lonafarnib inhibits RSV by acting on the RSV fusion protein rather than by inhibition of cellular farnesyltransferases. However, the logic of the article is not clear enough, the phenotypic and mechanistic study of lonafarnib on RSV are not deep enough, so the conclusions are not convincing. Overall, the present studies are preliminary in nature, the enthusiasm for this study is limited, and there are several important issues that need to be addressed.

Response: We thank this referee for the diligent analysis of our study and his/her constructive advice of how to amend the limitations of our study.

Major comment:

(1) In terms of mechanisms, the authors directly examined the effect of lonafarnib on RSV fusion protein (F protein) after excluding that lonafarnib did not affect RSV replication by inhibiting farnesyltransferase activity, and finally concluded that lonafarnib inhibited RSV infection by inhibiting F protein. This conclusion is less convincing. The authors should design experiments to verify which stage in the RSV life cycle is inhibited by lonafarnib (entry, uncoating, replication packaging, budding or release). For example, a time-of-addition assay that lonafarnib is added at different time points before and after RSV infection can be employed.

Response: We thank this referee for these constructive and clear suggestions, which helped us to extend the results showing by which mechanism lonafarnib inhibits RSV infection. To address this, we conducted several additional experiments. First, and as suggested by the referee, we performed a time of addition assay and included these new data as novel Figure 5A into our revised paper. These new data show that addition of lonafarnib has a very similar time-dependent inhibitory profile as ziresovir (a clinical stage F protein fusion inhibitor). For both compounds, co-administration of the drug together with the virus for 2h leads to the greatest inhibition. In contrast, for ribavirin, a replication inhibitor, highest drug potency is reached if it is added 2h post inoculation until 24h post inoculation.

Second, we conducted an RSV replicon assay. Using this assay, we are also able to show that unlike ribavirin, which dose-dependently inhibits RSV replication, lonafarnib does not show an inhibitory effect on RSV replicon. These results were included into the novel Figure 5B.

Third, we conducted an RSV F protein fusion assay. Specifically, we transfected 293T cells with a Venus-GFP expression construct and an RSV-F expression plasmids 6h prior to treatment of cells with the solvent control DMSO, 5 μ M Lonafarnib or 0.1 μ M ziresovir. Forty-eight hours later, we quantified the size of RSV F protein induced syncytia with an automated imaging work flow. The novel Figure 5C and the novel fluorescent microscopy images in Figure 5D show

that both lonafarnib and ziresovir significantly reduce the fusion efficiency of RSV F in this cell-based assay.

Collectively, these novel experimental data originating from the time of addition assay, the replicon assay and the RSV F protein fusion assay provide strong additional evidence for the conclusion that lonafarnib inhibits RSV infection through blocking the RSV F protein during membrane fusion. Further support of this conclusion is provided by the co-crystallization of lonafarnib with RSV F as described in our response to one of the following concerns (concern #4).

(2) The authors found that lonafarnib can directly bind to the fusion protein, but did not investigate the effect of lonafarnib on the function of the fusion protein. RSV fusion protein plays an important role in the viral entry and membrane fusion, so the authors can evaluate the effect of lonafarnib on virus-cell fusion and cell-cell fusion. For instance, the authors can detect the effect of lonafarnib on the syncytia formation. The known inhibitor of the RSV fusion protein, presatovir, could also be used as a positive control to perfect the experiments. Furthermore, before the RSV infection, the virions are first adsorbed to the cell surface and then initiate the virus-cell fusion. Thus, the authors can employ viral attachment experiments to assess whether lonafarnib blocks RSV attachment.

Response: Thank you for the suggestion. To further elucidate the molecular mechanism of lonafarnib, and as described above, we added an RSV-F fusion assay (Fig. 5CD). These new data provide evidence that lonafarnib inhibits RSV-F-induced membrane fusion, similar to ziresovir, a clinical-stage F protein inhibitor. Briefly, a plasmid encoding RSV-F was co-transfected with a GFP expressing plasmid into 293T cells. After transfection the cells were treated with lonafarnib in parallel with the positive control ziresovir, and the solvent control DMSO. Syncytia formation at the end point was visualized by fluorescence imaging and quantified by Fiji software. This quantitative imaging analysis of RSV-F-dependent fusion shows that addition of lonafarnib inhibits membrane fusion. Adding to this, the new results showing an antiviral activity of lonafarnib against recent clinical isolates also show reduced syncytia formation in presence of lonafarnib (compare the novel Figure 3B fluorescence imaging analysis). Finally, the novel results of the lonafarnib RSV F protein co-crystallization confirm the binding of lonafarnib to RSV F and provide a molecular framework for the inhibition of membrane fusion by lonafarnib (compare novel Figure 6).

(3) In the Figure 2, to assess the effect of lonafarnib on RSV infection, in addition to the Plaque assay, the authors should detect other indicators that can reflect RSV replication, such as the expression level of RSV P protein or N protein can be detected by Western blot, the mRNA level of RSV can be detected by the RT-qPCR or Northern blot.

Response: Thank you for this comment. In addition to the plaque assay, our dose-response curves in the new Fig. 3A was done by intracellular staining of RSV-P and flow cytometry detection. We were also able to show by immunofluorescence staining of RSV-P that lonafarnib could reduce RSV-F induced syncytia formation (Fig. 3B). Moreover, we used qRT-PCR based detection of RSV genomes for assessing the antiviral activity of lonafarnib *in vivo* and in differentiated BCI-NS1.1 cells.

(4) In the Figure 3E, The authors used the SPR assay to verify the interaction between fusion protein and lonafarnib. In order to make this conclusion more convincing, we believe that the interaction between lonafarnib and fusion protein mutant (K394R) should be supplemented here as a control and other experiments (such as ITC, BLI or CETSA) should be added to further verify the interaction. Moreover, if possible, the authors need to judge the specific binding sites between lonafarnib and fusion protein by the crystal structure analysis.

Response: We thank the referee for this suggestion and agree that additional work with the RSV F protein mutants would be of interest. However, we would like to point out that we provide firm phenotypic data showing that for instance the K394R exchange and the newly discovered T335I and T400A mutations cause resistance to lonafarnib. Considering this, we focused our attention on protein expression, purification and co-crystallization of the recombinant RSV F protein. In our view this was the most impactful way to confirm binding to F and to extend the molecular understanding of how lonafarnib inhibits RSV infection. Given this decision, we did not have access to the mutant RSV F proteins and were unable to complete these suggested experiments. Nevertheless, we believe that the data set presented in the revised paper makes a strong case for lonafarnib binding to RSV F and inhibiting cell entry and membrane fusion.

(5) In the Figure 4, the authors used RSV luciferase reporter virus or RSV GFP reporter virus to infect the immortalized lung cell lines and mouse model. We believe that there are differences between genetically modified viruses and natural RSV strains. Thus, the author should complement the natural RSV-infected cells and animal experiments. Moreover, in animal studies, the authors should increase the experimental regimen of dosing after infection as a control and add Ribavirin as a positive control to compare the antiviral effect with lonafarnib.

Response: Thank you for the comment. To address this concern, we have conducted several additional experiments and included these new results showing that recent clinical isolates of RSV are inhibited by lonafarnib. Please compare the novel Figure 3 highlighting the antiviral activity of lonafarnib against four recent RSV isolates (both A and B). Please also see the accompanying table 1 summarizing the IC₅₀ and IC₉₀ values. In addition, we used an untagged clinical isolate to establish the therapeutic effect of lonafarnib in BCI-NS1.1 cells (compare novel Figure 7D). Of note the IC₅₀/IC₉₀ values of the clinical RSV strains are comparable to the ones of the RSV A luciferase or GFP reporter viruses (compare table 1 and Figure 1C and 2A for RSV A luc and RSV A GFP, respectively).

Regarding the use of reporter viruses, we would like to point out that several research teams in the field use reporter viruses to test the efficacy of antivirals in vivo and that the system used by us was validated by multiple studies. In the paper by Rameix-Welti et al., 2017, we have shown that there is a perfect correlation between luciferase signal and viral mRNA by qRT-PCR. This was confirmed in Descamps et al., 2021 (<https://doi.org/10.1128/JVI.00912-21>). The RSV-Luc system is now largely validated and widely used and confirmation by RT-PCR has proven to be no longer necessary like in Jacque et al., 2021 (doi: 10.3389/fimmu.2021.683902); Riso-Balester et al., 2021 (<https://doi.org/10.1038/s41586-021-03703-z>), Palsson et al., 2020 (doi: 10.3389/fimmu.2020.580547), Galloux et al., 2020 (<https://doi.org/10.1128/AAC.00717-20>), Laubretton et al., 2020 (doi:10.3390/v12080822),

Blockus et al., 2020 (<https://doi.org/10.1016/j.antiviral.2020.104774>), Bryche et al., 2019 (<https://doi.org/10.1111/jnc.14936>), Gaillard et al., 2017 (<https://doi.org/10.1128/AAC.02241-16>), Cagno et al., 2017 (DOI: 10.1038/NMAT5053), Hervé et al., 2016 (<http://dx.doi.org/10.1016/j.jconrel.2016.10.003>). The recombinant virus has the same sequence as the Long strain except the insertion of a luciferase gene between P and M genes (Rameix-Welti et al., 2017).

We thank this referee also for the suggestion to conduct additional animal experiments aimed at extending the dosing regimens and at using ribavirin as positive control. We appreciate this suggestion, however, ultimately decided against conducting these additional animal experiments for ethical reasons. The two independent experiments conducted by us have reproducibly shown an antiviral effect of lonafarnib treatment at the level of both viral RNA genomes and reporter virus signal (Figure 8C and D). Therefore, a proof of concept for the *in vivo* antiviral activity had been established already. In our point of view, additional experimentation in animals would be justified if new approaches are available which are likely to enhance the potency of lonafarnib. We believe that this could be realized for instance by development of new formulations, which deliver high doses directly to the lumen of the lung and the mucus layer of lung epithelial cells (e.g. by inhalation). However, the design and testing of such novel formulations requires additional work, which in our view is beyond the scope of this present study. Given these considerations, we focused our attention on the other experimental approaches listed above and below to further improve our study.

Minor comment:

(1) Why the IC50 of lonafarnib differed 5-fold between the first round and second round of infection.

Response: We have observed this phenomenon for several other RSV entry inhibitors. In part this may be explained by the fact that compounds have two consecutive rounds of infections to unfold their antiviral potential. Factors like the half-life of the compounds under the assay conditions, may also influence this.

(2) In the Figure 2A-F, the authors should focus on the phenotypic studies of RSV inhibition by lonafarnib, while the results of the effect of Tipifarnib on RSV should be placed in the Supplementary Data.

Response: Thank you very much for the advice. After the addition of new results and figures, to keep in line with the flow of the story, we nevertheless decided to keep the effect of Tipifarnib together with lonafarnib in the manuscript. We prefer this way of presentation so that we contrast the two different inhibitors of the same cellular target in the same figure. We feel that this head-to-head presentation will make it easier for the reader to understand that lonafarnib likely does not inhibit RSV infection through acting on the host target but rather via inhibiting RSV F protein itself. We have added additional results in the revised Figures 3B, 5C and 5D depicting the phenotypic effect of lonafarnib on RSV F protein induced syncytia and cell-cell fusion.

(3) In the Figure 2G, the results of lonafarnib and Tipifarnib affecting HDV replication should be placed in the Supplementary Data.

Response: Thank you very much for the advice. After re-arranging the new figures and data, in line with the flow of the story, we nevertheless prefer to keep the results of lonafarnib and tipifarnib against HDV in the revised Fig. 2. We believe these data are important to validate that lonafarnib is antiviral and inhibits its known target (HDV).

(4) The discussion section of the article should be rewritten to focus on the development of RSV fusion protein inhibitors and studies related to lonafarnib.

Response: Thank you for your suggestion and we have revised the discussion section of the manuscript. We deleted some detail regarding the IMPDH and HSP90 inhibitors. Instead, we added some more discussion in regards to RSV fusion inhibitors and lonafarnib inducing its bioavailability.

Reviewer #3 (Remarks to the Author):

This manuscript describes the screening of a drug repurposing library for inhibitors of respiratory syncytial virus (RSV) infections. The authors identified a number of candidates which had been previously identified as inhibitors of various host cell proteins. The authors focused on one, lonafarnib, a farnesyltransferase inhibitor, as a potential inhibitor of RSV infections. The authors described inhibition of RSV infections in tissue culture cells and the toxicity of the drug to tissue culture cells. They showed that lonafarnib bound to purified soluble prefusion RSV F protein and that the inhibitor selected drug resistant mutations after 10 passages of RSV in tissue culture. Experiments in mice were presented to show efficacy of the inhibitor in vivo. Many of the experiments presented are moderately well done, particularly experiments done in tissue culture, and the results raise very interesting questions for future studies.

There are, however issues with the study that decrease enthusiasm for the development of this inhibitor for RSV.

1. The authors did not show inhibition of infections when the inhibitor was added after the virus. That is, they did not show time of addition of the inhibitor on inhibition relative to addition of virus in tissue culture.

Response: Thank you for pointing this out. We added a time of addition assay shown in the new Fig. 5A. From the result we can see that lonafarnib is most effective when co-incubated with the virus within 2 hours of virus infection, providing evidence that lonafarnib targets the early entry step of RSV. Similar concerns were raised by referees 1 and 2. Therefore, we conducted additional experiments to show that addition of lonafarnib after virus inoculation (i.e. in a therapeutic regimen) is antiviral.

First, we inoculated A549 cells with RSV-A-GFP at an MOI of 0.01. Twenty-four hours later, we washed the cells and added fresh medium containing antiviral compounds. Subsequently at 48h, 72h, 96h, 120h post inoculation (i.e. 24h up to 96h upon addition of compounds), we

determined RSV infection efficiency by measuring the number of infected cells using flow cytometry. Data were normalized to the highest number of infected cells as observed in DMSO control treated cells at 120h post inoculation. At this terminal timepoint and with therapeutic treatment of 5 μ M lonafarnib, RSV infected only ca. one third of the cell numbers infected in the DMSO-treated cells. This infection rate is somewhat lower compared to treatment with Ribavirin (100 μ M), which reduced infected cell number to ca. 50% as compared to the DMSO control (compare novel Figure 7B).

Second, we differentiated BCI-NS1.1 cells and cultured them in air liquid interface (ALI) configuration. Subsequently, we inoculated the cells with a recent clinical RSV-A strain (HRSV/A/DEU/H1/2013) at an MOI of 0.1 24h prior to treatment from the basolateral side. Apical washes were collected 72h and 96h post inoculation and a LDH toxicity analysis was performed from the basolateral media (lower graph). Viral genome equivalents (GE) were analyzed by qRT-PCR analysis. Under these conditions, 5 and 2.5 μ M lonafarnib reduced RSV genome equivalents ca. by 50% in the apical washes of treated BCI-NS1.1 ALI cultures. These novel data were included as novel Figure 7D. Collectively, these results provide evidence, that addition of lonafarnib after virus inoculation, i.e. therapeutic treatment, inhibits RSV infection and spread in A549 and differentiated BCI-NS1.1 cells.

2. In a related issue they only tested in mice the potential for prophylactic effect of inhibitor treatment but not it's therapeutic potential.

Response: We thank this referee for the suggestion to conduct additional animal experiments showing that lonafarnib has a therapeutic effect *in vivo*. We acknowledge this limitation of our data set. To address this, we conducted additional *in vitro* experimentation to document the therapeutic efficacy of lonafarnib in principle (compare new figures and description directly above). Given this new evidence, we ultimately decided against conducting these additional animal experiments primarily for ethical reasons. The two independent experiments conducted by us have reproducibly shown an antiviral effect of lonafarnib treatment at the level of both, viral RNA genomes and reporter virus signal. Therefore, a proof of concept for the *in vivo* antiviral activity (in a prophylactic setting) had been established already. We concur that it would be desirable to also show therapeutic efficacy *in vivo*. For this to work, it would be advantages to have means to improve drug exposure at the site of infection (the lumen of the lung and the epithelial cells). However, in our point of view, with the current modes of lonafarnib application it will be difficult to further boost drug exposure directly at the site of action (lumen of the lung). In our view, additional experimentation in animals would be justified once new approaches are available, which are likely to enhance the potency of lonafarnib. We believe that this could be realized for instance by development of new formulations, which deliver high doses directly to the lumen of the lung and the mucus layer of lung epithelial cells (e.g. by inhalation). However, the design and testing of such novel formulations requires additional work, which in our view is beyond the scope of this present study. Given these considerations, we focused our attention on the other experimental approaches listed above and below to further improve our study.

3. In mouse experiments, the authors used BalbC mice which are only semi permissive to RSV infection. Testing in more permissive animals, such as cotton rats, would be a more rigorous test of the potential of the inhibitor. Furthermore, in tests of efficacy in mice, the authors

added the inhibitor multiple times during the course of the experiment. Was this necessary to see an effect

Response: Thank you very much for your suggestion regarding our animal experiment. We are sorry to say that due to time and resource limits, testing RSV infection on cotton rats is out of reach for us. We added the inhibitor multiple times during the course of infection, because according to the PK/PD profile of lonafarnib in mice (supplementary Fig. S4), twice daily dosing can keep lonafarnib at a high and stable level in the plasma and lung. Presently, we do not know if the repetitive dosing is strictly required.

4. The legend or text does not describe the protocol used for results shown in Figure 4, panel J.

It appears that there is considerable lung pathology in drug treated animals, a result inconsistent with inhibition of infection in the animals.

Response: Thank you very much for pointing out this mistake. We have revised this figure to the new figure 8 in the revised manuscript, and we corrected the figure legend. It is true that, through HES staining, we see significant lung pathology in the first animal experiment compared with the second. The reasons for this finding are currently under investigation.

5. The authors did not show direct evidence that the drug blocked membrane fusion as would be expected if it inhibited cell entry, as the authors imply.

Response: Thank you for pointing this out. We did several additional experiments to address this important point. First, we added an RSV-F protein fusion assay (Fig. 5CD). These new data provide evidence that lonafarnib inhibits RSV-F-induced membrane fusion, similar to ziresovir, a clinical-stage F protein inhibitor. Briefly, a plasmid encoding RSV-F was co-transfected with a GFP expressing plasmid into 293T cells. After transfection the cells were treated with lonafarnib in parallel with the positive control ziresovir, and the solvent control DMSO. Syncytia formation at the end point was visualized by fluorescence imaging and quantified by Fiji software. This quantitative imaging analysis of RSV-F-dependent fusion shows that addition of lonafarnib inhibits membrane fusion. Adding to this, the new results showing an antiviral activity of lonafarnib against recent clinical isolates also show reduced syncytia formation in presence of lonafarnib (compare the novel Figure 3B fluorescence imaging analysis. Finally, the novel results of the lonafarnib RSV F protein co-crystallization confirm the binding of lonafarnib to RSV F and provide a molecular framework for the inhibition of membrane fusion by lonafarnib (compare novel Figure 6).

6. The conclusion that inhibition by lonafarnib is not due to its activity on farnesyltransferase is circumstantial and is premature without further experiments.

Response: We agree with this comment of this referee. We cannot strictly rule out that lonafarnib inhibits RSV infection also via acting on farnesyltransferases. We edited our discussion to make this point clear. Nevertheless, the extended data set provides strong experimental evidence that lonafarnib binds to the RSV F protein, and inhibits membrane fusion and infection (fusion assay (Fig. 5CD), SPR interaction study (Fig. 4F), co-crystallization (Fig. 6). Furthermore, considering that passaging of RSV in presence of lonafarnib selected for

resistance mutations within the RSV F protein, which confer phenotypic resistance, provide good evidence that binding to RSV F and inhibition of cell entry exerts selection pressure on the virus and is a key mechanism of the antiviral activity of Ionafarnib against RSV.

Other issues:

1. There is a lack of statistical analysis of some of the results. Examples are: Figure 2, panels H, C, F; Figure 3, panels C and D; Figure 4 panel C.

Response: Thank you very much for your suggestion. We have included statistical analysis in the revised figures 2C, 2E, 4C, 4D, 4E, 5B, 7B and 7E.

2. Some of the experiments are not well described and are difficult to follow. The figure legends or text should more completely describe protocols of experiments.

Response: Thank you very much for the advice. We have revised our old and new figure legends to describe the experiments in more detail.

3. The authors only used female animals.

Response: Historically, we decided to work with female animals because they fight less. As a result, they have fewer wounds and therefore less background. Furthermore, the sexual cycles of the females have no influence on either virus replication or the development of the weak pathological signs induced by RSV infection.

4. Figure 2, panel D bottom left, why is the fluorescence increasing with concentration of inhibitor?

Response: Thank you for pointing out this issue. The virus used here, the HRSV/A/DEU/H1/2013, is a fast progressing strain and can lead to high infectivity and cell toxicity at MOI of 1. Often in our experiments we observed that when doing dose-response curves for drug inhibitors, at high compound dose, the reduced virus infection results in increased cell viability, and even though the cells are infected, they are still viable, and thus results in a higher fluorescence intensity. As we do not observe this effect in other slow progressing strains of RSV B subgroup, to avoid confusion, we decided to remove the MFI in the revised Fig. 2 and Fig. 3.

Finally, we would like to thank all the referees and the editor for their careful and constructive evaluation of our study. Following your suggestions we were able to substantially improve our paper.

Reviewer #1 (Remarks to the Author):

The authors have responded adequately to the comments raised by the reviewers with new data. The revised data presented establishes lonafarnib as having antiviral activity against RSV by binding to the pre-F confirmation of the F protein and inhibiting its fusion activity to the host membrane. It will remain to be seen if the drug can be reformulated to improve its antiviral activity and reduce its toxicity.

Reviewer #3 (Remarks to the Author):

This manuscript is a revision of a submission describing the characterization of a compound, lonafarnib, identified in a screen of molecules identifying those that inhibited RSV infection in tissue culture cells.

There were a number of issues in the first submission raised by the reviewers. The authors' responses to some of them are satisfying but others are less so. As a result, the authors should temper or qualify some of their conclusions.

1. Reviewers called for assessment of therapeutic potential of the lonafarnib. The authors have argued that demonstration of an inhibition of spread of RSV infection in tissue culture cells addressed this property. However, infections in humans depends on the virus binding to the CX3CR1 in lungs, a molecule not present in tissue culture cells. Virus binding in tissue culture cells is largely due to F binding to GAGs. The role of this GAG binding in humans is not clear. Thus inhibition of spread in tissue culture does not directly test therapeutic potential in natural infections of humans (or the proper animal model). The results presented show only that the inhibitor, added multiple times, can block syncytia formation by progeny virus in tissue culture.

2. In a related issue, the authors have used syncytial formation as a measure of membrane fusion. Syncytia formation depends not only on membrane fusion, but also on binding of virus to cell receptors, pore expansion, uncoating of the virus, and cytoplasmic rearrangement. It is possible that lonafarnib blocks any of these steps. A direct test of membrane fusion would be content mixing or spread of lipids from effector to target membranes. Both assays are well established in the literature.

3. Data in Figure 8, particularly in panels D and G, are not particularly convincing of conclusions made in this manuscript. Mice are only semi-permissive to RSV infection (100 fold less permissive than cotton rats). Even so, the authors report minimal reduction in virus titer upon RSV challenge. Also they show significant lung pathology in one of two experiments after RSV challenge, a result inconsistent with the conclusion that lonafarnib blocks virus infection. The authors offer with no explanation.

4. Additionally, the syncytial in Figure 2, panels in D, Figure 3, panels in B, and Figure 5, panels in D, are not visible, thus the controls for these experiments are not clear and thus weaken conclusions of the experiments.

5. The authors use only female animals but do not clearly point this out. Conclusions that female reproductive cycles have no influence on results are not supported by a reference. For example, It is well known that females respond to immunization differently than males. There is some evidence of differences in response to RSV infections in animal models.

Reviewer #4 (Remarks to the Author):

Major point:

The PDB validation report shows the overall good quality of the structure. However, the mFo-DFc (at 3 rmsd) in purple (negative) and green (positive) in the report makes it seem the coordinates of some key atoms may be better improved.

Figure 6: One inhibitor molecule binds to each symmetry-related RSV-F protomer with an

occupancy of 0.33. Does the inhibitor itself have a three-fold symmetry?

Figure S5: The Br atoms probably should have a stronger electron density map compared to carbons, but it seems the Br does not show clear density at 2.5 sigma. Please explain.

Minor point:

Fig. 6 -> Figure 6

Reply to reviewer comments:

REVIEWER COMMENTS

Reviewer #1 (Remarks to the Author):

The authors have responded adequately to the comments raised by the reviewers with new data. The revised data presented establishes lonafarnib as having antiviral activity against RSV by binding to the pre-F confirmation of the F protein and inhibiting its fusion activity to the host membrane. It will remain to be seen if the drug can be reformulated to improve its antiviral activity and reduce its toxicity.

Answer: Thank you again for carefully reviewing our manuscript and for your helpful suggestions and comments. This has helped a lot to improve our manuscript.

Reviewer #3 (Remarks to the Author):

This manuscript is a revision of a submission describing the characterization of a compound, lonafarnib, identified in a screen of molecules identifying those that inhibited RSV infection in tissue culture cells.

There were a number of issues in the first submission raised by the reviewers. The authors' responses to some of them are satisfying but others are less so. As a result, the authors should temper or qualify some of their conclusions.

1. Reviewers called for assessment of therapeutic potential of the lonafarnib. The authors have argued that demonstration of an inhibition of spread of RSV infection in tissue culture cells addressed this property. However, infections in humans depends on the virus binding to the CX3CR1 in lungs, a molecule not present in tissue culture cells. Virus binding in tissue culture cells is largely due to F binding to GAGs. The role of this GAG binding in humans is not clear. Thus inhibition of spread in tissue culture does not directly test therapeutic potential in natural infections of humans (or the proper animal model). The results presented show only that the inhibitor, added multiple times, can block syncytia formation by progeny virus in tissue culture.

Answer: We concur with the notion of this referee and agree that *in vitro* infection experiments do not fully recapitulate the complexity of human infections. The referee is also correct with the statement that infection of human primary lung cells *ex vivo* depends on viral usage of CX3CR1. Likewise, infection of mice and cotton rats *in vivo* has been shown to depend on RSV G-protein CX3CR1 interactions (1, 2).

With our experiments in well-differentiated BciNS1.1 cells, which produce mucus and cilia, we tried to mimic the behavior of human primary lung cells (Fig. 7C and D). Our data in this model confirm prophylactic (Fig. 7C) and therapeutic efficacy (Fig. FD) of lonafarnib in this cellular system. However, while these Bci-NS1.1 cells are a well-accepted model of human primary lung cells, CX3CR1 expression and viral dependence on this molecule in this model has not been established. Therefore, to address this concern, we toned down our statements on the therapeutic efficacy of lonafarnib by pointing out these limitations of our study. Please compare our re-written text in the discussion section of our paper. "However, it should be noted that RSV infection of primary human lung cells *ex vivo* depends on C-X3-C motif chemokine receptor 1 (CX3CR1) (1). Likewise, mouse and cotton rat infection by RSV *in vivo* depend on CX3CR1 (1, 2). Given that it is currently not

known if the BCI-NS1.1 cell model recapitulates RSV-CX3CR1-dependence, caution is warranted when extrapolating these *in vitro* data to the complex viral receptor dependence in human lung cells.”

2. In a related issue, the authors have used syncytial formation as a measure of membrane fusion. Syncytia formation depends not only on membrane fusion, but also on binding of virus to cell receptors, pore expansion, uncoating of the virus, and cytoplasmic rearrangement. It is possible that lonafarnib blocks any of these steps. A direct test of membrane fusion would be content mixing or spread of lipids from effector to target membranes. Both assays are well established in the literature.

Answer: We agree with this comment of the referee and concur that syncytia formation in virus infected cell culture depends on multiple factors (as pointed out by the referee). To show that lonafarnib inhibits RSV F protein-dependent fusion we have transfected RSV F into 293T cells and measured syncytia formation by quantitative imaging (Figure 5C and 5D).

3. Data in Figure 8, particularly in panels D and G, are not particularly convincing of conclusions made in this manuscript. Mice are only semi-permissive to RSV infection (100 fold less permissive than cotton rats). Even so, the authors report minimal reduction in virus titer upon RSV challenge. Also they show significant lung pathology in one of two experiments after RSV challenge, a result inconsistent with the conclusion that lonafarnib blocks virus infection. The authors offer with no explanation.

Answer: In our view, our revised manuscript comments on these limitations. Please compare discussion section statements: “However, lonafarnib’s efficacy is lower compared with above mentioned clinical stage inhibitors (lonafarnib IC50 range against recent clinical strains from 10 to 118 nM compared to: riletamovir EC50 = 0.5nM; sisanatovir EC50 = 1.3nM, ziresovir EC50 = 5nM (3)). Furthermore, lonafarnib also inhibits farnesyltransferases and may therefore have unwanted side effects, particularly when administered orally and at high doses. This potential concern is stressed by our result showing that a 10-fold increased deposition of lonafarnib in the BALF correlated with enhanced antiviral activity but also side effects as plasma levels are close to the *in vitro* determined CC50 of lonafarnib. „

As mentioned in this sentence, we believe that high dose oral administration of lonafarnib reaches plasma levels close to the *in vitro* CC50. This could be the reason for side effects and limited potential as an antiviral.

We propose that alternative modes of administration may allow accumulation of lonafarnib at high doses in the lumen of the lung, while sparing other tissues and decreasing side effects:

“It is possible that alternative routes of lonafarnib administration improve the efficacy/side effect ratio. To explore this, the testing of alternative application routes and formulations could be useful. For instance, it is possible that inhalation of lonafarnib deposits high compound levels directly to the apical side of lung cells, where infection and cell to cell spread occurs. This route of administration may improve efficacy with a tolerable degree of side effects, because tissue-wide access to the host target, that is likely at least in part responsible for unwanted effects, may be reduced compared to the oral administration route.”

4. Additionally, the syncytial in Figure 2, panels in D, Figure 3, panels in B, and Figure 5, panels in D, are not visible, thus the controls for these experiments are not clear and thus weaken conclusions of the experiments.

Answer: Thank you for this comment, which provides us with the opportunity to clarify this issue. In Fig 2D, we analyzed the antiviral effect of lonafarnib on Hepatitis Delta Virus (HDV) infection. HDV does not induce syncytia, therefore the images do not show any syncytia. To further confirm an influence of lonafarnib on syncytia formation in RSV-infected cells we complement Fig 3B with an additional novel panel Fig. 3C. This novel panel 3C is a close-up view of RSV infected cells in presence or absence of lonafarnib. This new image shows the outline of cells and the nuclei helping identification of multi-nucleated syncytia. Similarly, we now complement Fig. 5C and D, which assess the influence of lonafarnib on F protein dependent syncytia formation in transfected 293T cells, with a magnified picture (novel Fig. 5E). Please find below one example, where we highlight the outline of one syncytium. As for Figure 5D and E, we included a magnification to make it easier for the reader to see the syncytium.

5. The authors use only female animals but do not clearly point this out. Conclusions that female reproductive cycles have no influence on results are not supported by a reference. For example, It is well known that females respond to immunization differently than males. There is some evidence of differences in response to RSV infections in animal models.

Answer: We agree with this comment and added a statement referring to this limitation in the discussion section of our re-revised paper: "Please note that only female animals were used in these *in vivo* experiments so that we cannot rule out gender specific effects on these results."

We thanks this referee for his/her expert advice, which has been very helpful to improve our manuscript.

Reviewer #4 (Remarks to the Author):

Major point:

The PDB validation report shows the overall good quality of the structure. However, the mFo-DFc (at 3 rmsd) in purple (negative) and green (positive) in the report makes it seem the coordinates of some key atoms may be better improved.

Answer: We understand the skepticism of the reviewer, as the conformation of lonafarnib does not in all aspects match the available density. Several aspects need to be considered to discuss the placement of the ligand:

1. The trimeric axis of the RSV F trimer follows a crystallographic symmetry axis of the trigonal spacegroup, implying that the ligand is positioned on a crystallographic symmetry axis. The inhibitor itself has no three-fold symmetry and sterically only one inhibitor can fit into each RSV F trimer, suggesting an estimated ligand occupancy of 0.33 / RSV F protomer in the experimental data. This implies that the density observed for lonafarnib is an averaged electron density. Together this might explain why the electron density in the ligand region is not well defined.
2. We therefore performed SAD phasing at the bromine peak as we considered the expected anomalous difference map for the bromine atoms our strongest guideline for placement of the ligand. In this map we observed only one strong, but slightly asymmetric peak (shown in Fig. S5) per RSV F protomer rather than the expected two peaks indicating the position of the two bromine atoms. This peak distribution clearly indicated the position of the first bromine atom and suggested that the second bromine atom is located at an almost symmetry-related position rotated around the three-fold axis (shown in Fig. S5). Based on this placement of the two bromine atoms the ligand was then placed in its entirety and refined against the full data, while maintaining both bromine atoms in the anomalous difference density.
3. Based on the restrictions explained above (placement of a non-symmetric inhibitor on a three-fold crystallographic axis) a better refinement was not feasible using state-of-the-art X-ray crystallography protocols.
4. We have addressed this issue by placing a more profound explanation of the strategy to place the Lonafarnib molecules in the methods of the modified manuscript: "As the placement of an asymmetric inhibitor on a 3-fold crystallographic symmetry axis was difficult, we combined the phases of the fully refined model with the anomalous differences derived from the two bromine atoms in the lonafarnib molecules to calculate an anomalous difference map. The major anomalous difference map peak revealed the position of the first bromine atom, and the fact that only a single anomalous difference map peak was observed together with the geometry of the ligand indicated that the second bromine atom was located on an almost symmetry-related position rotated around the 3-fold axis. Based on this placement of the two bromine atoms the intact inhibitor was placed."

Figure 6: One inhibitor molecule binds to each symmetry-related RSV-F protomer with an occupancy of 0.33. Does the inhibitor itself have a three-fold symmetry?

Answer: No, the inhibitor does not display any three-fold symmetry.

Figure S5: The Br atoms probably should have a stronger electron density map compared to carbons, but it seems the Br does not show clear density at 2.5 sigma. Please explain.

Answer: In the polder map contoured at 3.0 sigma (Figure 6) as well as in the $2mF_o-DF_c$ map in the validation report clear density can be observed for both Br atoms. We believe that for the reasons outlined above, the precision as well as the precise strength of the electron density map is not ideal, while still clear enough to unambiguously place the ligand.

Minor point:

Fig. 6 -> Figure 6

Answer: This abbreviation has been changed in the modified manuscript.

We thanks this referee for his/her expert advice, which has been very helpful to improve our manuscript.

References:

1. Johnson SM, McNally BA, Ioannidis I, Flano E, Teng MN, Oomens AG, Walsh EE, Peeples ME. 2015. Respiratory Syncytial Virus Uses CX3CR1 as a Receptor on Primary Human Airway Epithelial Cultures. *PLoS Pathog* 11:e1005318.
2. Green G, Johnson SM, Costello H, Brakel K, Harder O, Oomens AG, Peeples ME, Moulton HM, Niewiesk S. 2021. CX3CR1 Is a Receptor for Human Respiratory Syncytial Virus in Cotton Rats. *J Virol* 95:e0001021.
3. Groaz E, De Clercq E, Herdewijn P. 2021. Anno 2021: Which antivirals for the coming decade? *Annu Rep Med Chem* 57:49-107.

Reviewer #4 (Remarks to the Author):

Thank you for addressing the comments - no further questions.